# Reciprocal regulation of RIG-I and XRCC4 connects DNA repair with RIG-I immune signaling

Guijie Guo[1,2,5], Ming Gao[1,2,5], Xiaochen Gao[3,4], Bibo Zhu [3,4], Jinzhou Huang[1,2], Xinyi Tu[1,2], Wootae Kim[1,2], Fei Zhao[1,2], Qin Zhou[1,2], Shouhai Zhu[1,2], Zheming Wu[1,2], Yuanliang Yan[1,2], Yong Zhang[1,2], Xiangyu Zeng[1,2], Qian Zhu[1,2], Ping Yin[1,2], Kuntian Luo[1,2], Jie Sun [3,4], Min Deng [1,2✉] & Zhenkun Lou [1,2✉]

The RNA-sensing pathway contributes to type I interferon (IFN) production induced by DNA damaging agents. However, the potential involvement of RNA sensors in DNA repair is unknown. Here, we found that retinoic acid-inducible gene I (RIG-I), a key cytosolic RNA sensor that recognizes RNA virus and initiates the MAVS-IRF3-type I IFN signaling cascade, is recruited to double-stranded breaks (DSBs) and suppresses non-homologous end joining (NHEJ). Mechanistically, RIG-I interacts with XRCC4, and the RIG-I/XRCC4 interaction impedes the formation of XRCC4/LIG4/XLF complex at DSBs. High expression of RIG-I compromises DNA repair and sensitizes cancer cells to irradiation treatment. In contrast, depletion of RIG-I renders cells resistant to irradiation in vitro and in vivo. In addition, this mechanism suggests a protective role of RIG-I in hindering retrovirus integration into the host genome by suppressing the NHEJ pathway. Reciprocally, XRCC4, while suppressed for its DNA repair function, has a critical role in RIG-I immune signaling through RIG-I interaction. XRCC4 promotes RIG-I signaling by enhancing oligomerization and ubiquitination of RIG-I, thereby suppressing RNA virus replication in host cells. In vivo, silencing XRCC4 in mouse lung promotes influenza virus replication in mice and these mice display faster body weight loss, poorer survival, and a greater degree of lung injury caused by influenza virus infection. This reciprocal regulation of RIG-I and XRCC4 reveals a new function of RIG-I in suppressing DNA repair and virus integration into the host genome, and meanwhile endues XRCC4 with a crucial role in potentiating innate immune response, thereby helping host to prevail in the battle against virus.

[1] Department of Molecular Pharmacology and Experimental Therapeutics, Mayo Clinic, Rochester, MN, USA. [2] Department of Oncology, Mayo Clinic, Rochester, MN, USA. [3] Thoracic Diseases Research Unit, Division of Pulmonary and Critical Care Medicine, Department of Medicine, Mayo Clinic College of Medicine and Science, Rochester, MN, USA. [4] Department of Immunology, Mayo Clinic College of Medicine and Science, Rochester, MN, USA. [5] These authors contributed equally: Guijie Guo, Ming Gao. ✉email: deng.min@mayo.edu; lou.zhenkun@mayo.edu

The genome of a cell is constantly challenged by exogenous and endogenous DNA damaging agents such as irradiation (IR), carcinogens, and replication stress, which could threaten the integrity of genome and lead to a variety of diseases such as developmental defects, immune deficiency, and cancers[1,2]. To maintain genomic stability, cells have evolved a complicated DNA damage response (DDR) system, which is responsible for sensing DNA damage and repairing damaged DNA[3]. Double-strand breaks (DSBs), one of the most lethal types of DNA lesions in cells, are mainly repaired by homologous recombination (HR) and non-homologous end-joining (NHEJ) pathways[4,5]. HR is an error-free pathway, which requires an intact sister chromatid as the template in S/G2 phases, while NHEJ is highly error-prone, occurs throughout the cell cycle and directly ligases broken DNA ends in the absence of sequence homology[6,7].

Recent studies have linked genome instability to the innate immune response[8,9]. The cytosolic nucleic acid-sensing pathways, which are well known to mediate protective immune defenses against pathogen infection, could potentiate efficient antitumor immune responses. DNA damaging agents can cause the accumulation of DNA fragments or micronuclei in the cytosol that are sensed by the DNA sensor cyclic GMP–AMP synthase (cGAS). Once activated by cytosolic DNA, cGAS will catalyze the production of cyclic GMP–AMP, which then functions as a second messenger to activate downstream adapter protein stimulator of IFN genes (STING) and thereby initiating the STING/IRF3/type I interferon (IFN) signaling cascade[10,11]. The type I IFN-dependent innate and adaptive immunity contribute to the efficacy of radiotherapy[12,13]. Notably, several studies reported that inhibition of some chromatin regulators including lysine-specific histone demethylase 1 and DNA methyltransferase can trigger cytosolic sensing of double-stranded RNA (dsRNA) and induce a type I IFN response in different cancer cells[14–16]. Feng et al. proposed that both the cGAS/STING-dependent DNA-sensing pathway and the mitochondrial antiviral-signaling protein (MAVS)-dependent RNA-sensing pathway contribute to type I IFN production induced by IR to varying extent in different cell lines in the presence or absence of ATR inhibitors[17]. These data indicate that the RNA-sensing pathway is also critical for antitumor immunity in some circumstances. However, whether RNA sensors are involved in DNA repair is unknown.

Retinoic acid-inducible gene I (RIG-I), a key cytosolic viral RNA sensor, detects a broad range of viral RNAs, such as single-stranded RNA of negative or positive polarity and dsRNA[18–20]. RIG-I contains two N-terminal CARD domains (2CARD), a helicase domain and the C-terminal domain (CTD). In the absence of viral RNA, RIG-I is in an auto-inhibited state wherein 2CARD is repressed through intramolecular interaction with CTD. RNA binding via the helicase domain and CTD leads to the release of auto-repressed 2CARD, which then forms a tetramer and recruits MAVS through the CARD–CARD interaction. This interaction nucleates MAVS CARD filament formation, which in turn serves as a signaling platform to recruit downstream signaling molecules and thereby initiating a MAVS/IRF3/type I IFN signaling cascade[21]. The E3 ligase RIPLET is essential for RIG-I-mediated innate immune response[22]. RIPLET preferentially recognizes and ubiquitinates pre-oligomerized RIG-I on dsRNA. In addition, RIPLET can also cross-bridge RIG-I filaments on longer dsRNAs and amplify RIG-I immune signaling[23,24]. Interestingly, recent studies propose that nuclear-resident RIG-I can sense viral replication and induce antiviral immunity, suggesting a previously unrecognized subcellular milieu for RIG-I[25,26], which prompted us to explore the potentially functional significance of RIG-I in the nuclear.

In the current study, we found that RIG-I is recruited to double-stranded breaks (DSBs) and suppresses NHEJ. Mechanistically, RIG-I interacts with XRCC4 and impedes the formation of XRCC4/LIG4/XLF complex at DSBs. Moreover, we found that RIG-I hinders retrovirus integration into the host genome by suppressing NHEJ pathway, which is distinguished from its canonical role in suppressing RNA virus infection by initiating innate immune response. In addition, we found that XRCC4 has a critical role in RIG-I immune signaling and coordinates with RIG-I to suppress RNA virus replication in host cells by promoting efficient type I IFN response. Taken together, our findings reveal an inhibitory role of RIG-I in DNA repair and a crucial role of XRCC4 in RIG-I immune signaling. This reciprocal regulation of RIG-I and XRCC4 connects DNA repair with RIG-I immune signaling, and provides novel insights into complicated mechanisms underlying the virus and host interaction.

## Results

**RIG-I is recruited to DNA DSB sites and suppresses NHEJ**. To test whether RIG-I responds to DNA damage, human lung carcinoma cells (A549) and breast adenocarcinoma cells (MDA-MB-231) were treated with IR, and the intracellular localization of RIG-I was examined. IR treatment led to RIG-I accumulation in the chromatin fractions of treated cells without affecting the nuclear localization of RIG-I (Fig. 1a, b and Supplementary Fig. 1a). In addition, RIG-I agonist treatment induced expression of RIG-I, and a substantial amount of RIG-I localized to the chromatin (Supplementary Fig. 1b). Furthermore, we examined whether RIG-I is recruited to DNA damage sites. Chromatin immunoprecipitation (IP) assays demonstrated that RIG-I is recruited to site-specific DSB sites that are generated by *AsiSI* restriction enzyme[27] (Fig. 1c). We utilized a reporter system in U2OS cells to induce the DSB by FokI to examine the localization of RIG-I. Upon induction of the DSB, we found that RIG-I localized to the site of damage (Fig. 1d). In addition, RIG-I could also be recruited to laser-induced DNA damage sites following micro-IR (Supplementary Fig. 1c), suggesting the potential involvement of RIG-I in regulating DNA DSB repair.

HR and NHEJ are the two main repair pathways for DNA DSB[4]. To assess whether RIG-I is involved in DSB repair, HR and NHEJ reporter assays were performed. RIG-I overexpression significantly inhibited NHEJ but not HR (Fig. 1e, f and Supplementary Fig. 1d, e). In contrast, another RNA sensor MDA5 or the adapter protein MAVS had no effect on NHEJ (Supplementary Fig. 1f–i). Moreover, we evaluated the effect of RIG-I agonist on NHEJ, and found that both RIG-I agonists (3p-hpRNA, Poly: IC) treatment profoundly inhibited NHEJ but not HR (Supplementary Fig. 1j–n). RIG-I agonist-induced inhibition of NHEJ was reversed by RIG-I knockdown, suggesting the inhibition functions through RIG-I. Next, we examined the role of endogenous RIG-I on NHEJ repair. As shown in Fig. 1g, h and Supplementary Fig. 1o, p, RIG-I depletion led to an increase of NHEJ efficiency. Furthermore, we assessed the role of RIG-I on class switch recombination[28]. Compared with control cells, RIG-I knockdown cells showed an increase in class switch efficiency (Fig. 1i and Supplementary Fig. 1q), indicating that RIG-I depletion enhanced NHEJ.

To further confirm the functional involvement of RIG-I in regulating DSB repair, RIG-I knockdown or overexpressing cells were treated with IR and γH2AX foci, markers of DSB, were detected by immunofluorescence to determine whether RIG-I affects DDR and repair. As shown in Fig. 1j, k, less γH2AX foci were observed in RIG-I-depleted cells compared with that in control cells. In contrast, RIG-I overexpressing cells showed

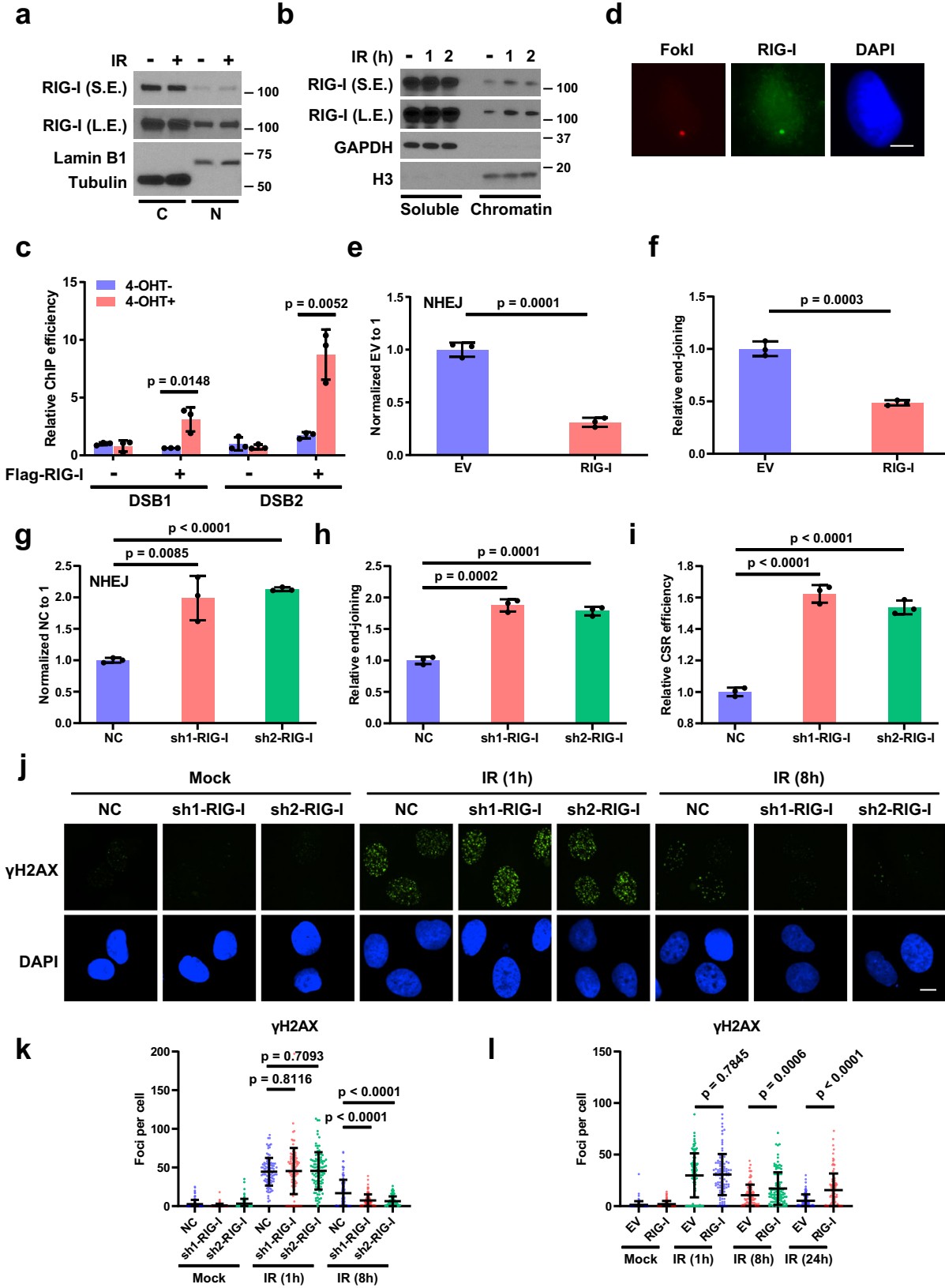

increased γH2AX foci at late time points (8 and 24 h) (Fig. 1l and Supplementary Fig. 1r). RIG-I agonist treatment also led to an increase of γH2AX foci at late time points in cells after IR treatment (Supplementary Fig. 1s, t), which suggests the important role of RIG-I in negatively regulating DNA repair.

Next, we asked how RIG-I is recruited to DSB sites. We examined the interaction of RIG-I with core NHEJ components (DNA-PK, Ku, XRCC4, LIG4, and XLF). Interestingly, RIG-I interacts with XRCC4 but not others, and the interaction increased in response to IR treatment and was highly RNA-

**Fig. 1 RIG-I is recruited to DNA DSB sites and suppresses non-homologous end-joining. a** A549 cells were treated with irradiation (IR, 10 Gy, 2 h). RIG-I protein levels in the cytosolic (C) and nuclear (N) fractions were detected by Western blot. **b** A549 cells were treated with IR (10 Gy) for the indicated times. RIG-I protein levels in the soluble and chromatin fractions were examined by Western blot. **c** ER-AsiSI U2OS cells were transfected with empty vector or Flag-RIG-I, and then treated with 4-OHT to induce DSBs. Flag-RIG-I accumulation at DNA damage sites generated by AsiSI was detected by ChIP-qPCR. Data are presented as mean values ± SEM from three independent experiments. *P* values are determined by unpaired two-sided *t*-test. **d** U2OS-FokI cells were treated with 1 mM Shield-1 and 1 mM 4-OHT for 5 h to induce site-specific double-strand breaks by FokI, and then fixed for immunofluorescence assay. RIG-I (green) localizes to the DSB site in U2OS-FokI cells where the DSB is induced by FokI (red). Scale bar, 10 μm. **e** Control and RIG-I overexpressing HEK293T cells were transfected with NHEJ reporter, and then cells were harvested for the NHEJ assay. Data are presented as mean values ± SEM from three independent experiments. *P* values are determined by unpaired two-sided *t*-test. **f** Control and RIG-I overexpressing HEK293T cells were transfected with linearized pEYFP plasmid for 12 h, followed by qPCR to detect the ligated EYFP region, normalized to an uncut flanking DNA sequence. Data are presented as mean values ± SEM from three independent experiments. *P* values are determined by unpaired two-sided *t*-test. **g** Control and RIG-I knockdown HEK293T cells were transfected with NHEJ reporter, and then cells were harvested for the NHEJ assay. Data are presented as mean values ± SEM from three independent experiments. *P* values are determined by unpaired two-sided *t*-test. **h** Control and RIG-I knockdown HEK293T cells were transfected with linearized pEYFP plasmid for 12 h, followed by qPCR to detect the ligated EYFP region, normalized to an uncut flanking DNA sequence. Data are presented as mean values ± SEM from three independent experiments. *P* values are determined by unpaired two-sided *t*-test. **i** Control and RIG-I knockdown CH12F3-2a cells were stimulated with ligands (TGF-β1, IL-4, and CD40 ligand), and class switch from IgM to IgA was analyzed. Data are presented as mean values ± SEM from three independent experiments. *P* values are determined by unpaired two-sided *t*-test. Representative pictures (**j**) and quantification (**k**) of γH2AX foci in control and RIG-I knockdown U2OS cells treated with IR (2 Gy) for the indicated time. Data are representative of three independent experiments. Each dot represents a single cell, and 100 cells were counted in each group for this experiment. Error bars represent ±SEM from this experiment. *P* values are determined by unpaired two-sided *t*-test. Scale bar, 10 μm. **l** Quantification of γH2AX foci in control and RIG-I overexpressing U2OS cells treated with IR (2 Gy) for the indicated time. Data are representative of three independent experiments. Each dot represents a single cell, and 100 cells were counted in each group for this experiment. Error bars represent ±SEM from this experiment. *P* values are determined by unpaired two-sided *t*-test.

dependent (Fig. 2a, b and Supplementary Fig. 2a). We generated two RIG-I mutants that are defective in binding RNA (K858, 861A; T347A)[29,30], and examined their interaction with XRCC4. As shown in Supplementary Fig. 2b, c, the interaction of wild type (WT), but not RNA binding-deficient RIG-I mutants, with XRCC4 was increased following IR treatment, suggesting that RNA binding is required for RIG-I to interact with XRCC4 following DNA damage, and XRCC4 may contribute to the recruitment of RIG-I to DSB sites. As expected, RIG-I localization to DSBs was dramatically decreased in XRCC4 knockdown cells (Fig. 2c and Supplementary Fig. 2d). Consistently, overexpression of WT, but not RNA binding-deficient RIG-I mutants whose interaction with XRCC4 cannot be efficiently induced following DNA damage, suppressed NHEJ (Supplementary Fig. 2e). On the other hand, MAVS, which is indispensable for RIG-I immune signaling, is not required for RIG-I-mediated inhibition of NHEJ (Supplementary Fig. 2f, g). In addition, overexpression of RIG-I would not further inhibit NHEJ in the absence of XRCC4 (Fig. 2d and Supplementary Fig. 2h). These results suggest that RIG-I is recruited to DSB sites in a XRCC4-dependent manner, and RIG-I overexpression suppresses NHEJ pathway through XRCC4.

**RIG-I suppresses NHEJ by disrupting the formation of XRCC4/LIG4/XLF complex at DSB sites.** We next wondered how RIG-I regulates NHEJ. Our results showed that the interaction of RIG-I and XRCC4 is significantly increased in response to DNA damage. Thus, we asked whether RIG-I regulates the formation and stability of XRCC4/LIG4/XLF complex at DSBs that are important for DNA repair by NHEJ. To test this hypothesis, we first checked the interaction of XRCC4 and LIG4 or XLF when RIG-I overexpressed. RIG-I overexpression or RIG-I agonists treatment markedly impeded the interaction of XRCC4 with LIG4 or XLF but not Ku80 induced by DNA damage (Fig. 3a and Supplementary Fig. 3a), suggesting that high expression of RIG-I affects the formation of XRCC4/LIG4/XLF complex but not the recruitment of XRCC4 by Ku80.

As expected, chromatin fractionation studies also revealed decreased chromatin binding of LIG4 and XLF but not Ku80 and XRCC4 in response to IR-induced DNA damage in RIG-I overexpressing cells (Fig. 3b and Supplementary Fig. 3b). In

addition, RIG-I agonists treatment also reduced the chromatin association of LIG4 and XLF, and the reduction is highly dependent on RIG-I (Supplementary Fig. 3c). Furthermore, we evaluated the effect of RIG-I on the localization of XRCC4/LIG4/XLF complex at DSB sites. As shown in Fig. 3c–e, RIG-I overexpression led to decreased localization of LIG4 and XLF but not XRCC4 at DSB sites.

We next mapped the domains of XRCC4 required for its interaction with RIG-I. As shown in Fig. 3f, both the N-terminal head domain and coil-coiled domain of XRCC4 that are responsible for the interaction between XRCC4 and XLF or LIG4 respectively[31–33], are required for its optimal interaction with RIG-I, suggesting that RIG-I competes with LIG4 and XLF to bind to XRCC4 (Fig. 3a and Supplementary Fig. 3a). These results could explain the reduced interaction of XRCC4 and LIG4/XLF in the presence of RIG-I. Furthermore, we mapped the domains of RIG-I responsible for XRCC4 interaction. As shown in Fig. 3g, deletion of the CTD of RIG-I dramatically impaired its interaction with XRCC4. Consistently, overexpression of RIG-I mutant lack of CTD had no effects on the interaction of XRCC4 and LIG4 or XLF, and NHEJ repair efficiency (Fig. 3h, i), suggesting that RIG-I suppresses NHEJ by interacting with XRCC4. Collectively, these results suggest that RIG-I suppresses NHEJ by interacting with XRCC4 and thereby disrupting the formation of the XRCC4/LIG4/XLF complex at DSB sites.

**The role of RIG-I in response to IR treatment.** We proceeded to assay the effect of RIG-I on IR sensitivity of cells. RIG-I overexpression sensitized cells to IR treatment (Fig. 4a and Supplementary Fig. 4a). Moreover, the IR hypersensitivity caused by RIG-I overexpression had not additional effect in the absence of XRCC4, suggesting that its role in regulating IR sensitivity is highly XRCC4-dependent (Fig. 4b and Supplementary Fig. 4b). A previous study suggests that RIG-I could regulate IR sensitivity through IFN signaling[34]. We found that high expression of RIG-I also promoted the sensitivity of IFNAR2 knockdown cells to IR treatment, indicating that in our experimental models, RIG-I regulates IR sensitivity in the absence of IFN-β signaling (Supplementary Fig. 4c–e). Next, we evaluated the effect of RIG-I agonist on IR sensitivity. As shown

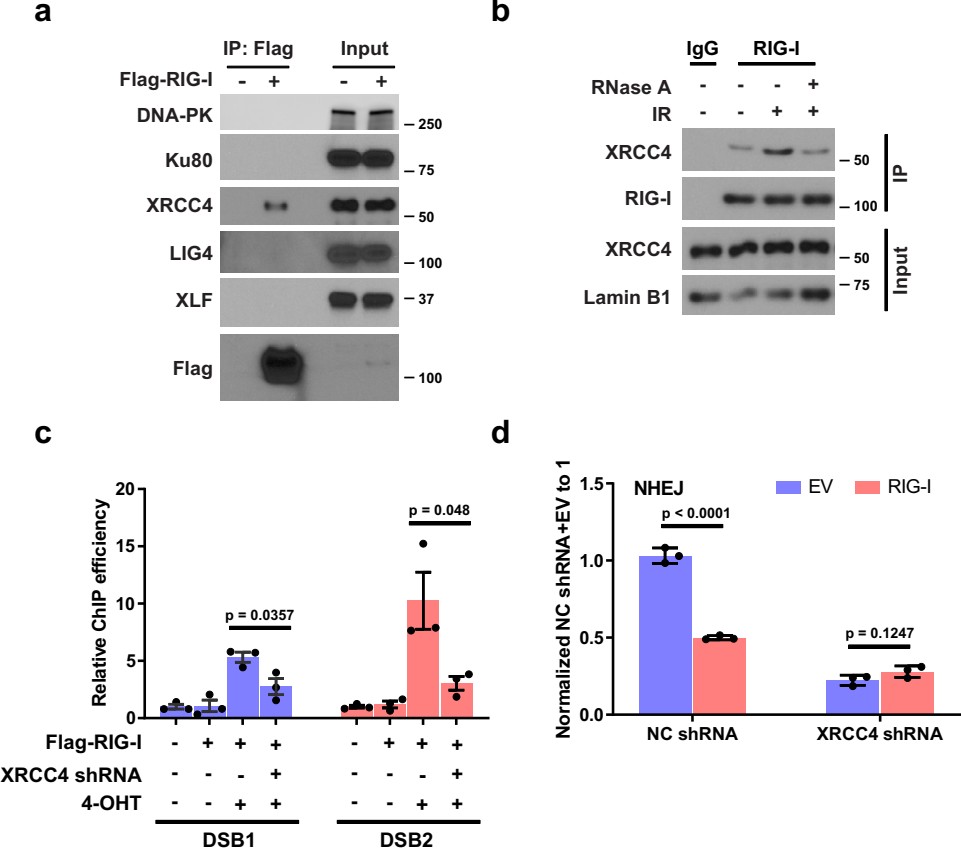

**Fig. 2 XRCC4 is required for the recruitment of RIG-I to DSB sites. a** HEK293T cells were transfected with empty vector or Flag-RIG-I. The cells were then lysed and immunoprecipitated with anti-Flag agarose beads. The beads were boiled and probed with indicated antibodies. **b** A549 cells were treated with IR (10 Gy, 1–2 h). Cells were lysed, and nuclear fractions were immunoprecipitated with anti-RIG-I antibody. The beads were treated with RNase A, boiled and blotted with indicated antibodies. **c** Control and XRCC4 knockdown ER-AsiSI U2OS cells were transfected with Flag-RIG-I, and then treated with 4-OHT to induce DSBs. Flag-RIG-I accumulation at DNA damage sites generated by AsiSI was detected by ChIP-qPCR. Data are presented as mean values ± SEM from three independent experiments. *P* values are determined by unpaired two-sided *t*-test. **d** Control and XRCC4 knockdown HEK293T cells overexpressing RIG-I were transfected with NHEJ reporter, and then cells were harvested for the NHEJ assay. Data are presented as mean values ± SEM from three independent experiments. *P* values are determined by unpaired two-sided *t*-test.

in Supplementary Fig. 4f, RIG-I agonist treatment sensitized cells to IR treatment, and loss of RIG-I in treated cells was able to reverse the increased IR sensitivity caused by RIG-I agonist treatment, demonstrating that RIG-I agonist regulated IR sensitivity in a RIG-I-dependent manner. In addition, RIG-I agonist treatment could not further sensitize the XRCC4 knockdown cells to IR treatment (Supplementary Fig. 4g), revealing the involvement of XRCC4 in RIG-I agonist-mediated IR hypersensitivity. Our results demonstrate that high expression of RIG-I or RIG-I agonist sensitizes cancer cells to IR treatment by suppressing NHEJ pathway.

We next examined the role of endogenous RIG-I on cellular sensitivity to IR. Control and RIG-I knockdown or knockout cells were treated with IR, and cell survival was analyzed by colony-formation assay. As shown in Fig. 4c, d and Supplementary Fig. 4h, i, RIG-I depletion rendered cells more resistant to IR treatment. Consistently, increased NHEJ efficiency and less γH2AX foci were observed in RIG-I-depleted cells (Supplementary Fig. 4j–o), suggesting that loss of RIG-I rendered cells more resistant to IR treatment through promotion of NHEJ. Furthermore, we examined the role of RIG-I on cellular sensitivity to IR in vivo. As shown in Fig. 4e and Supplementary Fig. 4p, RIG-I depletion rendered tumors more resistant to IR in the xenograft model. These results suggest that loss

of RIG-I rendered tumors more resistant to IR through promotion of NHEJ.

**RIG-I suppresses retrovirus integration into the host genome by impeding NHEJ.** NHEJ plays a critical role in retrovirus integration into the host genome[35]. It is possible that RIG-I, in addition to its well-established role in antiviral immunity, has a role in blocking viral integration into the host genome. Thus we tested whether RIG-I regulates retrovirus integration into the genome. We infected RIG-I overexpressing cells with GFP-positive lentivirus and examined GFP levels in the genomic DNA. As shown in Fig. 4f and Supplementary Fig. 4q, less GFP gene levels and provirus copies were detected in the genomic DNA extracted from RIG-I overexpressing cells than those from control cells, suggesting that RIG-I impedes retrovirus integration into genome. Conversely, MAVS, the adapter protein indispensable for RIG-I immune signaling, was not required for RIG-I-mediated inhibition of viral integration (Supplementary Fig. 4r). RIG-I agonist treatment also led to reduced retrovirus integration (Supplementary Fig. 4s, t). We evaluated the viral infection efficiency by quantifying GFP RNA levels in cells during early infection with the lentivirus (2 h). As shown in Supplementary

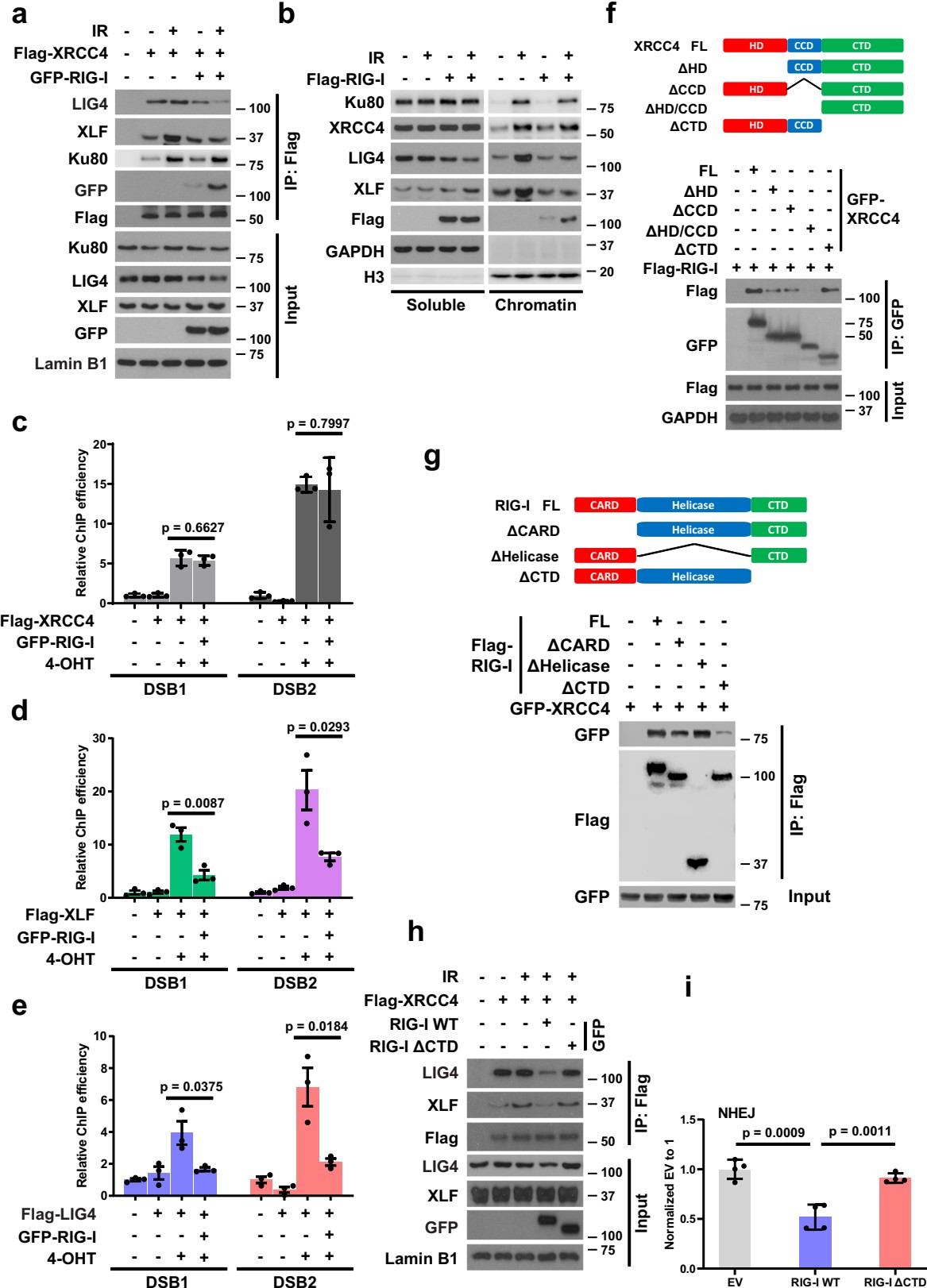

Fig. 4u, comparable GFP RNA levels were observed in 3p-hpRNA-treated and -untreated cells. In addition, neither RIG-I overexpression nor depletion had a significant effect on viral infection into host cells (Supplementary Fig. 4v, w), suggesting that it is the viral integration into host genome but not initial viral infection into host cells that was regulated by RIG-I. In addition, we found that the inhibitory effect of RIG-I in regulating retrovirus integration was not observed in the absence of XRCC4 (Fig. 4g), suggesting that RIG-I regulates retrovirus integration in a XRCC4-dependent manner. Furthermore, we found that

**Fig. 3 RIG-I suppresses non-homologous end-joining by disrupting the formation of XRCC4/LIG4/XLF complex at DSB sites. a** HEK293T cells were transfected with Flag-XRCC4 and GFP-RIG-I, and then treated with IR (10 Gy, 1 h). The cells were lysed and immunoprecipitated with anti-Flag agarose beads. The beads were boiled, subjected to SDS-PAGE, and analyzed with indicated antibodies. **b** HEK293T cells were transfected with Flag-RIG-I, and then treated with IR (10 Gy, 1 h). The soluble and chromatin fractions were separated, subjected to SDS-PAGE, and analyzed with indicated antibodies. **c–e** ER-AsiSI U2OS cells were transfected with Flag-XRCC4, Flag-XLF, or Flag-LIG4, and then treated with 4-OHT to induce DSBs. Flag-XRCC4 (**c**), Flag-XLF (**d**), and Flag-LIG4 (**e**) accumulation at DNA damage sites generated by AsiSI was detected by ChIP-qPCR. Data are presented as mean values ± SEM from three independent experiments. *P* values are determined by unpaired two-sided *t*-test. **f** Schematic representation of XRCC4 constructs used in this study (top). HEK293T cells were transfected with indicated GFP-XRCC4 constructs and Flag-RIG-I. Cell lysates were incubated with anti-GFP agarose beads. The immunoprecipitates were blotted with indicated antibodies (bottom). **g** Schematic representation of RIG-I constructs used in this study (top). HEK293T cells were transfected with indicated Flag-RIG-I constructs and GFP-XRCC4. Cell lysates were incubated with anti-Flag agarose beads. The immunoprecipitates were blotted with indicated antibodies (bottom). **h** HEK293T cells were transfected with Flag-XRCC4 and wild type (WT) or RIG-I mutant lack of C-terminal domain (ΔCTD), and then treated with IR (10 Gy, 1–2 h). The cells were lysed and immunoprecipitated with anti-Flag agarose beads. The beads were boiled, subjected to SDS-PAGE, and analyzed with indicated antibodies. **i** Control and WT or RIG-I mutant (ΔCTD) overexpressing HEK293T cells were transfected with NHEJ reporter, and then cells were harvested for the NHEJ assay. Data are presented as mean values ± SEM from three independent experiments. *P* values are determined by unpaired two-sided *t*-test.

---

overexpression of WT, but not the RIG-I mutant unable to interact with XRCC4, hinders retrovirus integration (Fig. 4h), indicating that RIG-I impedes retrovirus integration by interacting with XRCC4 and thereby suppressing NHEJ. Next, we examined the role of endogenous RIG-I on viral integration. As shown in Fig. 4i–k and Supplementary Fig. 4x, higher genomic GFP DNA levels and more provirus copies were observed in RIG-I-depleted MEF cells compared with that in control cells, suggesting that loss of RIG-I promoted viral integration into the host genome.

Overall, our results reveal the inhibitory role of RIG-I in regulating retrovirus integration into the host genome by impeding NHEJ pathway, which is distinguished from its canonical role in suppressing RNA virus infection by initiating innate immune response. This dual functions of RIG-I could enhance antiviral activities.

**Loss of XRCC4 attenuates RIG-I immune signaling.** While being essential for DNA repair, DDR proteins have been shown to participate in the sensing of intracellular foreign DNA, which results in the initiation of an innate immune response[36–41]. Having shown that RIG-I interacts with XRCC4 and regulates NHEJ pathway, we asked whether XRCC4 may also regulate RIG-I immune signaling that is essential for suppressing RNA virus infection in host cells. To test this hypothesis, we first examined the interaction of XRCC4 and RIG-I in the cytosol. Interestingly, cytosolic XRCC4 interacts with RIG-I, and the interaction was markedly increased in response to RIG-I agonist treatment and was highly RNA-dependent (Fig. 5a and Supplementary Fig. 5a).

To test whether XRCC4 responds to RIG-I signaling, cells were treated with RIG-I agonist, and the intracellular localization of XRCC4 was examined. As shown in Fig. 5b, c, RIG-I agonist treatment led to robust accumulation of XRCC4 in the mitochondrial fractions of treated cells, which displayed highly concurrent localization with RIG-I, implying the potential involvement of XRCC4 in RIG-I immune signaling. To further investigate the function of XRCC4 in RIG-I signaling, XRCC4 knockdown cells were challenged with RIG-I agonists, and IRF3 phosphorylation and IFN-β levels detected. As shown in Fig. 5d and Supplementary Fig. 5b–f, XRCC4 deficiency but not inactivation of DNA-PK led to dramatic reduction of IRF3 phosphorylation and IFN-β levels, and the reduction is highly dependent on RIG-I that no further reduction was observed when XRCC4 silenced in the absence of RIG-I (Fig. 5e and Supplementary Fig. 5g–i).

Next, we wondered how XRCC4 regulates RIG-I signaling. RIPLET has a critical role in RIG-I immune signaling[19,22].

RIPLET recognizes pre-assembled RIG-I oligomers on dsRNA and ubiquitinates RIG-I. The oligomerization of RIG-I is a prerequisite as RIPLET binds RIG-I only in complex with dsRNA that accommodates at least two RIG-I molecules[23,42]. It is well known that XRCC4 exists predominantly as a dimeric form, and can form homo-multimers as well as higher order structures or filaments[31]. Thus we asked whether XRCC4 may regulate the oligomerization of RIG-I by interacting with and tethering RIG-I. To test this, Flag- and GFP-tagged RIG-I were co-transfected into 293T cells followed by IP. As shown in Fig. 5f and Supplementary Fig. 5j, much less GFP-RIG-I was brought down by Flag-RIG-I in XRCC4 knockdown cells than that in control cells, suggesting that depletion of XRCC4 impaired the oligomerization and assembly of RIG-I on RIG-I agonists. Consistently, loss of XRCC4 led to reduced RIG-I ubiquitination due to the insufficient oligmerized RIG-I that cannot be efficiently recognized and ubiquitinated by RIPLET (Fig. 5g). In addition, the association of RIG-I with MAVS, which is highly dependent on the ubiquitination and oligomerization of RIG-I, was impaired in the absence of XRCC4 (Fig. 5h and Supplementary Fig. 5k). Furthermore, we found that re-overexpression of WT, but not the XRCC4 mutant devoid of interaction with RIG-I, could reverse the insufficient oligomerization of RIG-I and impaired interaction of RIG-I and MAVS caused by loss of XRCC4 (Fig. 5f, h), indicating that XRCC4 contributes to RIG-I signaling by interacting with RIG-I and promoting oligomerization and ubiquitination of RIG-I. Consistent with these results, overexpression of WT, but not the XRCC4 mutant devoid of interaction with RIG-I, could rescue the reduced IFN-β levels cause by depletion of XRCC4 (Fig. 5i). Collectively, our results reveal the important role of XRCC4 in regulating RIG-I immune signaling, and suggest that XRCC4 may coordinate with RIG-I to restrict RNA virus infection.

**XRCC4 coordinates with RIG-I to suppress RNA virus replication in host cells.** To further confirm the crucial role of XRCC4 in RIG-I signaling, we next examined the effect of XRCC4 on virus replication in host cells. We infected XRCC4 knockdown cells with influenza virus and monitored virus replication. As shown in Fig. 6a and Supplementary Fig. 6a, much more virus copies were detected in XRCC4 knockdown cells than that in control cells, while DNA-PK inhibitor had no such effect. In addition, the observed increase of viruses caused by loss of XRCC4 can be reversed by overexpression of WT but not the XRCC4 mutant devoid of interaction with RIG-I (Fig. 6b). In line with these results, dramatic decrease of IFN-β and interferon-stimulated gene (ISG) levels was observed in XRCC4 knockdown cells infected with influenza virus, which can be rescued by

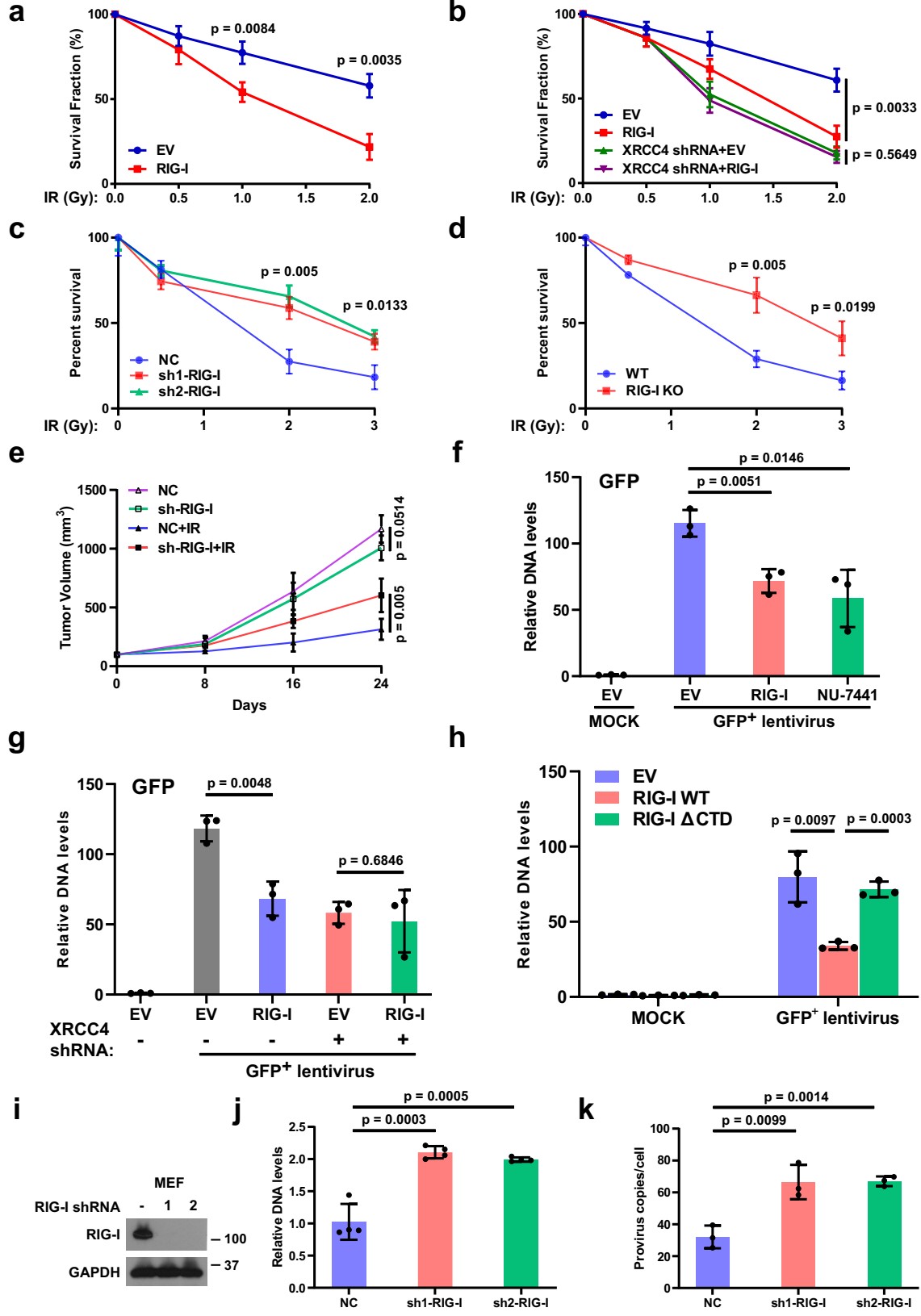

overexpression of WT but not the XRCC4 mutant unable to interact with RIG-I (Fig. 6c, d and Supplementary Fig. 6b–c). Our results demonstrate that XRCC4 coordinates with RIG-I to suppress RNA virus replication by promoting efficient production of type I IFN in host cells.

Next, we examined the role of XRCC4 on influenza virus replication in vivo. C57BL/6 mice were inhaled aerosolized siRNA targeting XRCC4 or control siRNA mixed with in vivo-jetPEI® using an air-compressing nebulizer (at an estimated inhalable dose of 3 mg/kg)[43–47]. After 24 h, mice were inoculated intranasally

**Fig. 4 RIG-I suppresses retrovirus integration into the host genome by impeding non-homologous end-joining. a** Colony-formation assay for control and RIG-I overexpressing A549 cells exposed to different dosage of irradiation (IR). Data shown are representative of three independent experiments and values are mean ± SEM of technical replicates ($n = 3$). $P$ values are determined by unpaired two-sided $t$-test. **b** Control and XRCC4 knockdown A549 cells overexpressing RIG-I were treated with indicated dosage of IR. Cell viability was assessed using the colony-formation assay. Data shown are representative of three independent experiments and values are mean ± SEM of technical replicates ($n = 3$). $P$ values are determined by unpaired two-sided $t$-test. Cell survival analysis of RIG-I knockdown (**c**) or knockout (**d**) A549 cells treated with IR. Data shown are representative of three independent experiments and values are mean ± SEM of technical replicates ($n = 3$). $P$ values are determined by unpaired two-sided $t$-test. **e** Control or RIG-I knockdown A549 cells were subcutaneously injected into the flank of NOD-SCID mice. Mice were treated with or without IR. Tumor volumes were monitored. Data points represent (mean ± SEM) are shown from $n = 5$ biologically independent samples by two-sided unpaired $t$ test. **f** HEK293T cells were transfected with Flag-RIG-I or treated with DNA-PK inhibitor (NU-7441, 2 μM, 24 h). The cells were then infected with GFP-positive lentiviruses. Genomic DNA was extracted. GFP levels in the genomic DNA were analyzed by qPCR. Data are presented as mean values ± SEM from three independent experiments. $P$ values are determined by unpaired two-sided $t$-test. **g** Control and XRCC4 knockdown HEK293T cells were transfected with Flag-RIG-I, and then infected with GFP-positive lentiviruses. GFP levels in the genomic DNA were analyzed by qPCR. Data are presented as mean values ± SEM from three independent experiments. $P$ values are determined by unpaired two-sided $t$-test. **h** HEK293T cells were transfected with wild type (WT) or RIG-I mutant lack of C-terminal domain (ΔCTD), and then infected with GFP-positive lentiviruses. GFP levels in the genomic DNA were analyzed by qPCR. Data are presented as mean values ± SEM from three independent experiments. $P$ values are determined by unpaired two-sided $t$-test. **i, j** Control and RIG-I knockdown MEF cells (**i**) were infected with GFP-positive lentiviruses. GFP levels in the genomic DNA were analyzed by qPCR (**j**). Data (**j**) are presented as mean values ± SEM from four independent experiments. $P$ values are determined by unpaired two-sided $t$-test. **k** Control or RIG-I knockdown MEF cells were infected with GFP-positive lentiviruses. Provirus copies in the genomic DNA were analyzed by qPCR. Data are presented as mean values ± SEM from three independent experiments. $P$ values are determined by unpaired two-sided $t$-test.

with $10^3$ PFU of influenza virus A/WSN/1933. Further treatment of siRNA was performed at 24 and 48 h after virus infection, respectively. As shown in Supplementary Fig. 6d, XRCC4 siRNA significantly downregulated XRCC4 expression in lungs of mice, compared with that in lungs of control mice. Lung XRCC4 knockdown mice displayed faster body weight loss and poorer survival than control mice during viral infection (Fig. 6e, f). In addition, lung XRCC4 knockdown mice exhibited a greater degree of lung injury caused by influenza virus infection than control mice (Fig. 6g and Supplementary Fig. 6e), suggesting that silencing XRCC4 in lung promotes influenza virus replication in vivo. Consistent with these observations, higher viral NP RNA levels and lower IFN beta RNA levels were observed in lung tissues with XRCC4 knockdown, compared with that in control tissues (Fig. 6h and Supplementary Fig. 6f). These results indicate that XRCC4 depletion renders mice more susceptible to influenza virus infection.

Taken together, our results reveal an inhibitory role of RIG-I in NHEJ pathway and thereby regulating retrovirus integration into the host genome, which is distinguished from its canonical role in suppressing RNA virus infection by initiating innate immune response. Reciprocally, XRCC4 plays a critical role in RIG-I immune signaling and coordinates with RIG-I to suppress RNA virus replication in host cells by promoting efficient type I IFN production. This reciprocal regulation of RIG-I and XRCC4 extends the antiviral functions of RIG-I, and endues XRCC4 with a crucial role in potentiating innate immune response, thereby helping the host to prevail in the battle between host and virus.

## Discussion

Recognition of DNA damaging agents-induced DNA fragments or micronuclei by DNA sensors links genome instability to innate immune response[34]. The RNA-sensing pathway also contributes to type I IFN signaling induced by DNA damaging agents[17,25]. However, the potential involvement of RNA sensors in DNA repair remains unknown. Here, we found that RIG-I, a key viral RNA sensor that recognizes RNA virus and activates innate immune response by initiating the MAVS/IRF3/type I IFN signaling cascade[48], is recruited to DSBs and suppresses NHEJ. Mechanistically, RIG-I interacts with XRCC4, and the RIG-I/XRCC4 interaction impedes the formation of XRCC4/LIG4/XLF complex at DSBs. Furthermore, high expression of RIG-I

sensitizes cancer cells to IR treatment. In addition, we also evaluated the effect of RIG-I on cellular sensitivity to IR in vivo, and found that loss of RIG-I rendered tumors resistant to IR in xenograft models. Collectively, our findings suggest that RIG-I may represent a potential target for cancer therapy, and provide the basis for the use of RIG-I agonist as a radio-sensitization treatment for cancers. Interestingly, cGAS, a cytosolic DNA sensor that senses DNA virus and activates immune response by initiating the STING/IRF3/type I IFN cascade, has recently been shown to suppress HR by disrupting the formation of the PARP1–Timeless complex[49]. These observations suggest a unique role of host immune signaling in regulating DNA repair, and raise a possibility that cancer cells with activated innate immunity could be more readily killed with DNA damage-inducing therapies, and DNA/RNA sensors may be employed as a unique biomarker in cancer therapy independent of their role in antitumor immunity.

RNA viruses are a diverse group of pathogens that are responsible for some prevalent and lethal human diseases including cancers. RNA viruses can induce robust DNA damage, which contributes to the viral pathogenesis through introduction of deleterious mutations that could increase the risk of tumorigenesis[50–52]. During RNA virus infection, cytosolic RIG-I can be employed by host to initiate innate immune response to fight against RNA virus infection and protect host. Our studies also suggest that nuclear RIG-I may be hijacked by virus to suppress the NHEJ pathway, promote genome instability and increase the risk of tumorigenesis. This might reveal an unexpected involvement of RIG-I in the cancer induced by RNA viruses.

However, that is quite another matter for retrovirus infection. NHEJ pathway plays a critical role in retrovirus integration into the host genome[35,53,54]. Therefore, the inhibitory role of RIG-I in regulating NHEJ pathway could be a host protective mechanism to restrict retrovirus infection by hindering retrovirus integration into the host genome. Notably, neither RIG-I overexpression nor depletion had an obvious effect on initial viral infection into host cells. Therefore, this mechanism is distinguished from its canonical role in suppressing RNA virus infection by initiating innate immune response. Although other steps from initial infection to eventual viral genome integration could still be affected by RIG-I, we think its inhibitory role on XRCC4 and NHEJ is a major factor, given the key role of NHEJ in viral integration into the

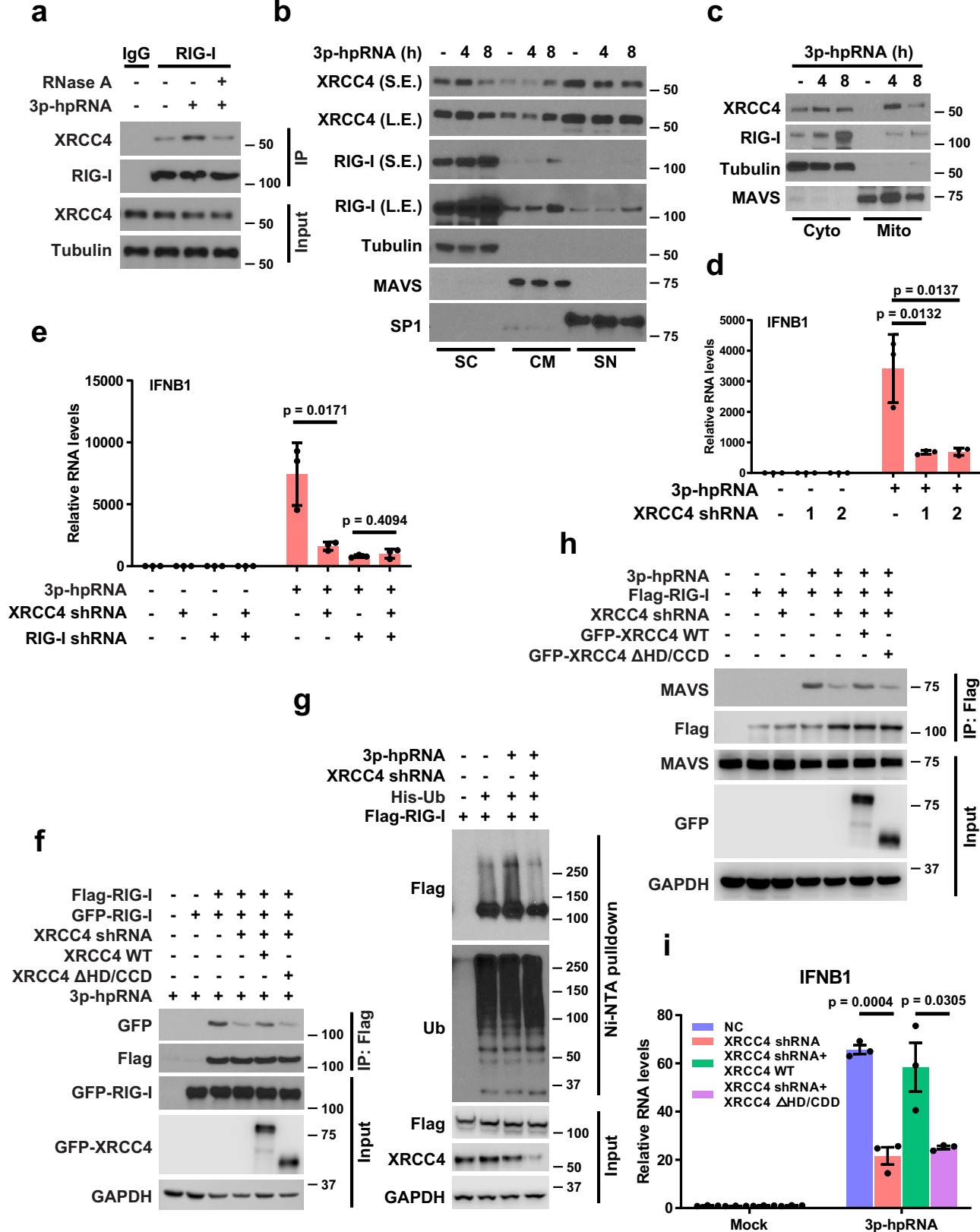

genome. Our study raises the possibility that RIG-I agonist or peptides could be used to hinder retrovirus infection by impeding retrovirus integration into the host genome.

Our study also suggests a new role of XRCC4 in RIG-I immune signaling. XRCC4 contributes to RIG-I immune signaling by regulating the oligomerization and ubiquitination of RIG-I, which led to efficient type I IFN response and thereby restricting RNA virus replication in host cells. Furthermore, in vivo studies showed that disrupting XRCC4 expression in lung promoted IAV replication in mice. Lung XRCC4 knockdown mice exhibited high

**Fig. 5 Loss of XRCC4 attenuates RIG-I immune signaling. a** A549 cells were transfected with 3p-hpRNA (0.5 μg/ml, 8 h). The cells were lysed, and cytosolic fractions were immunoprecipitated with anti-RIG-I antibody. The beads were treated with RNase A, boiled and blotted with indicated antibodies. **b** A549 cells were transfected with 3p-hpRNA (0.5 μg/ml) for the indicated time. Cells were subjected to subcellular fractionation into soluble cytoplasmic (SC), cytoplasmic membranous (CM), and soluble nuclear (SN) components. Western blot was performed with indicated antibodies. **c** A549 cells were transfected with 3p-hpRNA (0.5 μg/ml) for the indicated time. Cells were subjected to fractionation into cytosolic (Cyto) and mitochondria (Mito) components. Western blot was performed with indicated antibodies. **d** IFN-β RNA levels in control and XRCC4 knockdown A549 cells transfected with 3p-hpRNA (0.5 μg/ml, 12 h) were analyzed by qRT-PCR. Data are presented as mean values ± SEM from three independent experiments. *P* values are determined by unpaired two-sided *t*-test. **e** Control and XRCC4 knockdown A549 cells expressing RIG-I shRNA were transfected with 3p-hpRNA (0.5 μg/ml, 12 h). IFN-β RNA levels were detected by qRT-PCR. Data are presented as mean values ± SEM from three independent experiments. *P* values are determined by unpaired two-sided *t*-test. **f** XRCC4 knockdown HEK293T cells re-expressing wild type (WT) or XRCC4 mutant lack of the head and coil-coiled domains (ΔHD/CCD) were transfected with Flag- and GFP-tagged RIG-I and then 3p-hpRNA (0.5 μg/ml, 8 h). The cells were lysed, and immunoprecipitated with anti-Flag agarose beads. The beads were boiled and blotted with indicated antibodies. **g** Control and XRCC4 knockdown HEK293T cells were transfected with Flag-RIG-I and His-Ub and then 3p-hpRNA (0.5 μg/ml, 8 h). Cell lysates were immunoprecipitated with Ni-NTA (His) beads, and then blots were probed with indicated antibodies. **h** XRCC4 knockdown cells re-expressing wild type (WT) or XRCC4 mutant (ΔHD/CCD) were transfected with Flag-RIG-I and then 3p-hpRNA (0.5 μg/ml, 8 h). The cells were lysed, and immunoprecipitated with anti-Flag agarose beads. The beads were boiled and blotted with indicated antibodies. **i** XRCC4 knockdown cells re-expressing WT or XRCC4 mutant (ΔHD/CCD) were transfected with 3p-hpRNA (0.5 μg/ml, 12 h). IFN-β RNA levels were detected by qPCR. Data are presented as mean values ± SEM from three independent experiments. *P* values are determined by unpaired two-sided *t*-test.

susceptibility to IAV infection, as evidenced by enhanced lung injury and consequently decreased survival rates, suggesting that XRCC4 acts as a critical host factor and coordinates with RIG-I to defend against IAV infection. Under the circumstance of RNA virus infection, canonical function of nuclear XRCC4 in NHEJ pathway is suppressed by activated or induced expression of RIG-I. This might restrict viral integration into the host genome. On the other hand, XRCC4 is entrusted by host to interact and coordinate with RIG-I to promote efficient type I IFN production and fights against RNA virus infection. These suggest a dual role of XRCC4 in the involvement of RIG-I-mediated suppression of NHEJ and virus integration, and simultaneously in the enhancement of RIG-I-mediated protective type I IFN response, representing an ingenious response against the viral interference of host DNA repair in the tug of war between the host and virus.

Collectively, our findings reveal an inhibitory role of RIG-I in DNA repair and a crucial role of XRCC4 in RIG-I immune signaling, and provide novel insights into complicated mechanisms underlying the reciprocal regulation between virus and host.

## Methods

**Cell culture**. HEK293T, A549, and U2OS cell lines were purchased from ATCC. All of the cell lines have been tested and confirmed by the Mayo Clinic medical genome facility Center. U2OS ER-AsiSI cells were generated by Dr Gaëlle Legube (Université de Toulouse, Toulouse, France). HEK293T and A549 cells were maintained in DMEM, and U2OS cells were cultured with McCoy's 5A with 10% FBS.

**Virus infection**. Influenza virus strain A/PR/8/34 (H1N1) was prepared as previously described[55]. For infection, cells were washed with phosphate-buffered saline (PBS) and infected with A/PR/8/34 (H1N1). After 1-h adsorption, cells were washed once with warm PBS and cultured with DMEM containing 1% FBS for the indicated time.

**Plasmids, reagents, and antibodies**. RIG-I, MDA5, MAVS, XRCC4, LIG4, and XLF cDNA were purchased from Addgene, and subcloned into lentiviral vectors. pEYFP-N1 was kindly provided by Yi Sun (Department of Radiation Oncology, University of Michigan, Ann Arbor, MI)[32]. lentiCRISPR v2-hRIG-I sgRNA was kindly provided by Junjie Chen (The University of Texas MD Anderson Cancer Center, Houston, TX)[56]. His-tagged ubiquitin was obtained from Addgene[57]. Flag agarose beads (A2220) were purchased from Sigma-Aldrich. 3p-hpRNA and Poly: IC were purchased from Invivogen. RIG-I (3743, 1:1000), MAVS (3993, 1:1000), and phospho-IRF3 (4947, 1:1000) antibodies were purchased from Cell Signaling Technology. XLF (ab33499, 1:1000) antibody was purchased from Abcam. XRCC4 (sc-271087, 1:1000), GFP (sc-9996, 1:1000), and Ub antibodies (sc-8017, 1:1000) were purchased from Santa Cruz. DNA-PK (MA5-13238, 1:2000) and Ku80 (MA5-12933, 1:2000) antibodies were purchased from Thermo Fisher. Flag (F3165, 1:1000) antibody was purchased from Sigma-Aldrich. Lamin B1 (12987-1-AP, 1:2000), LIG4 (12695-1-AP, 1:500), and GAPDH (60004-1-Ig, 1:2000) antibodies were purchased from Proteintech. Alexa Fluor® 488 AffiniPure Donkey Anti-Rabbit IgG (H + L 715-585-150), Alexa Fluor® 594 AffiniPure Donkey Anti-Rabbit

IgG (H + L 711-585-152), Alexa Fluor® 488 AffiniPure Donkey Anti-Mouse IgG (H + L 715-545-150), and Alexa Fluor® 594 AffiniPure Donkey Anti-Mouse IgG (H + L 715-585-150) were purchased from Jackson Lab. Donkey Anti-Mouse IgG (H + L) ML (715-675-151) and Donkey Anti-Rabbit IgG (H + L) ML (711-675-152) were purchased from Jackson ImmunoResearch.

**DNA transfection, viral packaging, and lentiviral infection**. DNA transfections were performed using TransIT-X2 (MIRUS Bio). Lentiviruses were packaged in HEK293T cells in which indicated constructs, pMD2.G (Addgene), and pSPAX2 (Addgene) were co-transfected. Media containing lentiviruses were collected 48 h after transfection. Harvested media were added to infect cells for further experiments.

**RNA interference**. The following shRNAs from Sigma-Aldrich were used in this study:

human RIG-I shRNA-1: 5′-AGCACTTGTGGACGCTTTAAA-3′,
human RIG-I shRNA-2: 5′-CCAGAATTATCCCAACCGATA-3′,
mouse RIG-I shRNA-1: 5′-ACTGGAACAGGTCGTTTATAA-3′,
mouse RIG-I shRNA-2: 5′-GCCATGCAACATATCTGTAAA-3′,
human XRCC4 shRNA-1: 5′-TGTGTGAGTGCTAAGGAAGCT-3′,
human XRCC4 shRNA-2: 5′-CCTCAGGAGAATCAGCTTCAA -3′.

**Quantitative PCR**. RNA was isolated with TRIzol RNA Isolation Reagents (Thermo Fisher). Reverse transcription was performed with PrimeScript™ RT Reagent Kit (Takara), and quantitative PCR was performed with Power SYBR Green Master Mix (Thermo Fisher). For quantification, the $2^{-\Delta\Delta Ct}$ method was used to calculate the relative RNA levels against GAPDH. The provirus content in cells that have been transduced with lentivirus was detected using Lenti-X provirus quantitation kit (631239, Takara) according to the manufacturer's instruction. The copy number of proviruses in cells was determined by detecting integrated proviruses in genomic DNA using real-time quantitative PCR. The sequences of qPCR primers are shown in Supplementary Table 1.

**Chromatin fractionation**. Chromatin fractionation was performed as described previously[5]. In brief, cells were harvested and resuspended in low salt buffer (10-mM Tris-HCl, pH 7.4, 0.2-mM MgCl₂, 50-mM β-glycerophosphate, 10-mM NaF, and 1 mg mL⁻¹ each of pepstatin A and aprotinin) containing 1% Triton X-100 on ice for 15 min. After centrifugation (14,000 × g, 10 min), the supernatant contained the soluble proteins, and the pellet contained the chromatin-bound proteins. The pellets were then resuspended in 0.2-N HCl on ice for 15 min, sonicated, and neutralized with 1 M Tris-HCl, pH 8.0. The fractions were analyzed by SDS-PAGE and Western blot.

**Subcellular fractionation**. Subcellular fractionation and mitochondria isolation were performed using the subcellular protein fractionation kit (78840, Thermo Fisher) and mitochondria isolation kit (89874, Thermo Fisher), respectively, according to the manufacturer's instruction. The fractions were subjected to SDS-PAGE and Western blot.

**Western blot and IP**. Cells were harvested and lysed with NETN buffer (20-mM Tris-HCl, pH 8.0, 100-mM NaCl, 1-mM EDTA, 0.5% Nonidet P-40 with 10-mM NaF, and 1 mg per ml each of pepstatin A and aprotinin). After centrifugation at 12,000 × g for 15 min, supernatant containing proteins were immunoprecipitated by indicated antibodies or agarose beads overnight at 4 °C. The

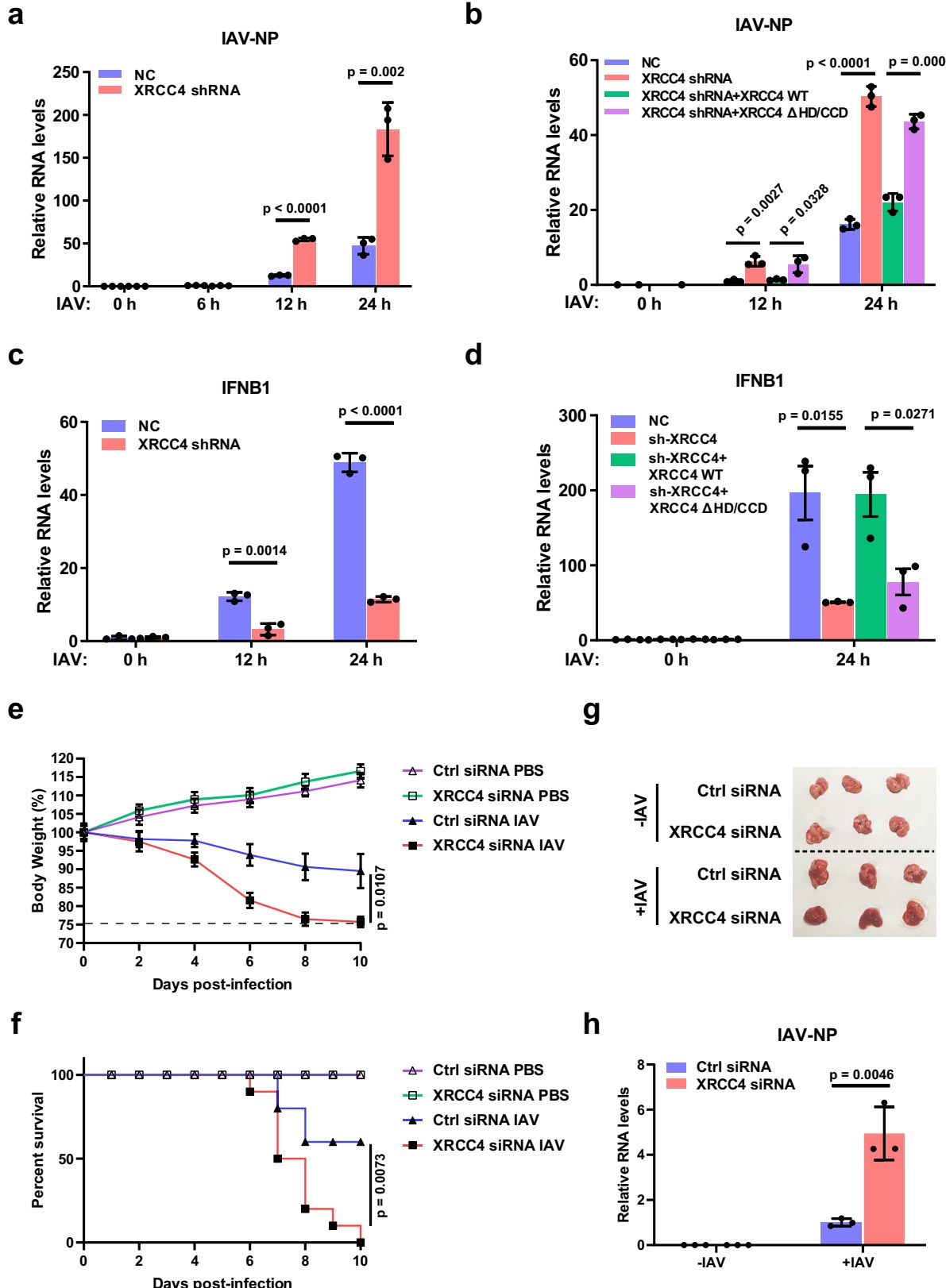

immunoprecipitates were washed with NETN buffer, centrifuged at $800 \times g$ for 1 min for three to five times. The immunoprecipitates were suspended with Laemmli buffer and boiled for SDS-PAGE.

**ChIP-qPCR**. The ChIP assay was performed using a Simple ChIP Enzymatic Chromatin IP kit (Cell Signaling Technology) following the manufacturer's

protocol. Briefly, The ER-AsiSI U2OS cells were treated with 4-hydroxytamoxifen (4-OHT; Sigma-Aldrich) to induce DSBs[27]. Next, cells were cross-linked with 1% formaldehyde and neutralized with 125-mM glycine. The cross-linked nuclear lysates were digested with micrococcal nuclease, and then sonicated to yield genomic DNA fragments between 150 and 900 bp. The digested chromatin was immunoprecipitated with the indicated primary antibody overnight at 4 °C. The immunocomplexes were pulled down with magnetic beads, reversely cross-linked

**Fig. 6 XRCC4 coordinates with RIG-I to suppress RNA virus replication in host cells. a** Control and XRCC4 knockdown A549 cells were infected with influenza virus A/PR/8/34 (IAV-PR8) as indicated. IAV NP RNA levels were detected by qRT-PCR. Data are presented as mean values ± SEM from three independent experiments. *P* values are determined by unpaired two-sided *t*-test. **b** IAV NP RNA levels in XRCC4 knockdown cells re-expressing wild type (WT) or XRCC4 mutant lack of the head and coil-coiled domains (ΔHD/CCD) infected with IAV-PR8 as indicated were examined by qRT-PCR. Data are presented as mean values ± SEM from three independent experiments. *P* values are determined by unpaired two-sided *t*-test. **c** Control or XRCC4 knockdown A549 cells were infected with IAV-PR8 as indicated. IFN-β RNA levels were analyzed by qRT-PCR. Data are presented as mean values ± SEM from three independent experiments. *P* values are determined by unpaired two-sided *t*-test. **d** IFN-β RNA levels in XRCC4 knockdown cells re-expressing WT or XRCC4 mutant (ΔHD/CCD) infected with IAV-PR8 as indicated were detected by qRT-PCR. Data are presented as mean values ± SEM from three independent experiments. *P* values are determined by unpaired two-sided *t*-test. **e, f** C57BL/6 mice were inhaled aerosolized control or XRCC4 siRNA for 24 h followed by infection with influenza virus A/WSN/1933 (IAV-WSN, $10^3$ PFU) (8–10 mice/group). The effects of XRCC4 on WSN virulence and infection kinetics in mice were determined by body weight loss (**e**) and cumulative survival curve (**f**). Body weight was measured every 2 days. Error bars represent ±SEM from this experiment. The dashed line in **e** indicates the endpoint of 25% weight loss. *P* values are determined by unpaired two-sided *t*-test (**e**) or one-sided log-rank [Mantel–Cox] test (**f**). **g** C57BL/6 mice were treated with control or XRCC4 siRNA followed by infection with IAV-WSN. Shown are representative pictures of lung tissues with or without WSN infection. **h** IAV NP RNA levels in the lung tissues of mice treated with control or XRCC4 siRNA followed by infection with IAV-WSN were detected by qRT-PCR. Data are presented as mean values ± SEM from three independent experiments. *P* values are determined by unpaired two-sided *t*-test.

at 65 °C for 30 min, and digested with proteinase K overnight. The DNA samples were purified using Miniprep Columns. Real-time PCR was performed with StepOne Plus Real-Time PCR System (Applied Biosystems) using SYBR Select Master Mix (Applied Biosystems). The sequences of qPCR primers are shown in Supplementary Table 1.

**FokI system**. The U2OS-FokI cells contain a stably integrated LacO array and stably express the mCherry-LacI-FokI fusion protein fused to a destabilization domain (DD), and a modified estradiol receptor (ER) (ER-mCherry-LacI-FokI-DD) was kindly provided by Roger A. Greenberg (Department of Pathology, University of Pennsylvania, Philadelphia, PA). To induce site-specific DSBs by FokI, cells were treated with 1-mM Shield-1 and 1-mM 4-OHT for 5 h[58]. Cells were then fixed for immunofluorescence assay.

**HR and NHEJ reporter assay**. Cells were transfected with DR-GFP or EJ5, pCBA-I-SceI, and pCherry. After 2 days, cells were harvested and analyzed by flow cytometry to examine the percentage of GFP-positive cells. Results were normalized to control group. The graphical account for FACS sequential gating/sorting strategies was provided in Supplementary Fig. 7.

**Class switch recombination**. Class switch recombination was performed in CH12F3-2a cells[28]. Briefly, cells were stimulated with ligands (1 ng/ml of recombinant human TGF-β1, 10 ng/ml of recombinant murine IL-4, and 250-ng/ml recombinant murine CD40 ligand). After 60 h, cells were collected, stained with murine IgA antibody (eBiosciences, 12-5994-82) and murine IgM antibody (eBiosciences; 11-5890-82) to analyze class switch from IgM to IgA. Cells were analyzed on a FACS Calibur (BD Biosciences). Results were normalized to control group. The graphical account for FACS sequential gating/sorting strategies was provided in Supplementary Fig. 7.

**NHEJ linearized plasmid assay**. The linearized plasmid-based end-joining assay was performed as described previously[32]. Briefly, pEYFP-N1 was linearized by digesting with NheI. The linear products were transfected into serum-starved cells. Cells were then harvested and lysed to isolate transfected plasmids. The efficiency of end-joining repair was assessed by qPCR of the ligated YFP region, normalized to an uncut flanking DNA sequence. Results were normalized to control group. The sequences of qPCR primers are shown in Supplementary Table 1.

**Immunofluorescence staining**. Cells were seeded on coverslips for 24 h before experiments. For detection of γH2AX, cells were fixed by 4% (w/v) paraformaldehyde for 15 min at room temperature following by washing with PBS, and then permeabilized for 5 min at room temperature with 0.5% (v/v) Triton X-100 followed by washing with PBS. For detection of RIG-I, cells were treated with CSK buffer (10-mM HEPES-KOH, pH 7.4, 300-mM sucrose, 100-mM NaCl, 3-mM MgCl₂) containing 0.5% (v/v) Triton X-100 for 5 min on ice, and then washed with CSK buffer lacking Triton X-100 and fixed with 4% (w/v) paraformaldehyde for 30 min. Cells were blocked with 5% goat serum for 30 min and then incubated with primary antibodies at 4 °C overnight. After washing with PBS, secondary antibody was added and incubated for 1 h at room temperature before stained with 4′6-diamidino-2-phenylindole. The coverslips were mounted onto glass slides with anti-fade solution and visualized using a Nikon ECLIPSE E800 fluorescence microscope. The foci intensity was quantified with Image J software.

**Denaturing Ni-NTA pull-down**. Cells were harvested and lysed in Urea buffer (8-M Urea, 0.1-M NaH₂PO4, 0.01-M Tris-HCl pH 8.0, and 300-mM NaCl). Lysates were then sonicated and incubated with Ni-NTA beads for 2 h at room temperature. After washing the beads with Urea wash buffer (8 M Urea, 0.1 M NaH₂PO4, 0.01 M Tris-HCl pH 8.0, and 300 mM NaCl) for at least five times, the immunocomplexes were suspended with Laemmli buffer and subjected to SDS-PAGE and immunoblotting.

**Colony-formation assay**. A549 cells were plated in each well of six-well plates and then treated with IR as indicated. After incubated for 12–14 days at 37 °C, colonies were stained with 5% GIEMSA and counted. The survival of untreated cells was normalized to 100%.

**Tumor xenograft**. Animal experiments were performed under the approval of the Institutional Animal Care and Use Committee at Mayo Clinic (Rochester, MN). All mice used in this study were maintained under specific pathogen-free conditions, 21 ± 2 °C relative humidity of 45 ± 15%, and a 12-h light/dark cycle. Control or RIG-I knockdown A549 cells ($5 \times 10^6$) were injected subcutaneously into the flanks of 6-week-old female athymic nude Ncr nu/nu mice (National Cancer Institute/National Institutes of Health). When the tumor volume reached around 100 mm³, IR was applied locally to the tumor-bearing legs of mice at 10 Gy. Tumor volume was measured every 8 days and calculated using the formula (length × width²)/2. Mice were sacrificed for tumor dissection 24 days after the start of treatment.

**Mouse infection**. Animal experiments were performed under the approval of the Institutional Animal Care and Use Committee at Mayo Clinic (Rochester, MN). All mice used in this study were maintained under specific conditions, 21 ± 2 °C relative humidity of 45 ± 15%, and a 12-h light/dark cycle. Female C57BL/6 mice (5-6 weeks old) were inhaled aerosolized siRNA targeting XRCC4 or control siRNA mixed with in vivo-jetPEI® (Polyplus) using an air-compressing nebulizer (at an estimated inhalable dose of 3 mg/kg). After 24 h, mice were inoculated intranasally with $10^3$ PFU of influenza virus A/WSN/1933 (H1N1). After 24 and 48 h, mice were retreated with control or XRCC4 siRNA, respectively. Mice were weighted every 2 days for a period of 10 days. Viral NP RNA levels and IFN beta RNA levels in lungs were quantified by qRT-PCR. The sequences of qPCR primers are shown in Supplementary Table 1.

**Statistics and reproducibility**. Data in bar and line graphs are presented as mean ± SEM of at least three independent experiments. Western blotting and micrograph data were repeated independently three times with similar results. Data shown in the figures are representative of three biologically independent experiments. For the animal xenograft study, data are presented as the mean ± SEM of five mice. Statistical analyses were performed in Microsoft Excel 2010 and GraphPad Prism 8 with the Student's two-tailed *t* test. The flow cytometry data were collected using Attune NxT Flow Cytometer software v2.6 and analyzed by flowjo V10.

**Reporting summary**. Further information on research design is available in the Nature Research Reporting Summary linked to this article.

## Data availability
A reporting summary for this article is available as a Supplementary Information file. All data are available from the authors upon reasonable request. Source data are provided with this paper.

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

## Acknowledgements

We thank Dr Junjie Chen (The University of Texas MD Anderson Cancer Center, Houston, TX) for providing lentiCRISPR v2-hRIG-I sgRNA. We thank Dr Roger A. Greenberg (Department of Pathology, University of Pennsylvania, Philadelphia, PA) for providing U2OS-FokI cells. We thank Dr Yi Sun (Department of Radiation Oncology, University of Michigan, Ann Arbor, MI) for providing pEYFP-N1. We thank members of Lou lab for comment and discussion throughout the project. This work was supported by funding from NIH RO1 AG047156, RO1 AI112844, and RO1 AI147394 (J.S.), and CA 203971 (Z.L.).

## Author contributions

G.G. and M.G. designed and conducted experiments, analyzed data, and wrote the paper. M.D., X.G., B.Z., J.H., X.T., W.K., F.Z., Q. Zhou, S.Z., Z.W., Y.Y., Y.Z., X.Z., Q. Zhu, K.L. and J.S. provided technical and data analysis assistance. P.Y. provided reagents for experiments. Z.L. and M.D. conceived and supervised the project, designed experiments, and analyzed data.

## Competing interests

The authors declare no competing interests.
