## [Peer Review File · Nature Communications]

REVIEWER COMMENTS

Reviewer #1 (Remarks to the Author):

In this manuscript entitled "Reciprocal regulation of RIG-I and XRCC4 connects DNA repair with RIG-I immune signaling" described the connection between immune signaling and double-stranded break (DSB) repair. They found that RIG-I, a key cytosolic RNA sensor for immune signaling, interacts with the NHEJ key factor XRCC4. This interaction impedes the interaction of XRCC4 with other NHEJ proteins, and thus suppresses NHEJ, which is required for retrovirus integration. In the other hand, XRCC4 contributes RIG-I immune signaling. Thus, these findings reveal the mechanism how RIG-I and NHEJ coordinate to inhibit virus infection.

Overall, this is a very interesting story and contains a lot of high quality data. Thus, I am exceptionally supportive of this work and recommend its publication in Nature Communications.

Some comments:

1. Figure 1A-D, RIG-I is recruited to DNA damage sites using fractionation and CHIP. Can this be confirmed by IR- or Laser-induced foci.
2. The NHEJ reporter system used in this paper is very crude. The results will strongly effected by many factors, such as transfection efficiency, cell status after lacking essential gene, and so on. Therefore, a stringent assay for NHEJ is necessary, such as class switch recombination (CSR) analysis, telomere fusion analysis in TRF2-depleted cells, or foreign DNA integration analysis.
3. The stability of LIG4 depends on its interaction with XRCC4. When RIG-I depletes XRCC4 from LIG4 after it's overexpressed or cells are treated by RIG-I agonist, is the protein level of XRCC4 reduced?
4. Where is the level of non-phosphorylated IRF3 in Figure S5B-D, S5F-H?

Reviewer #2 (Remarks to the Author):

In their manuscript, Guo and co-workers study the role of RIG-I on the cellular response to DNA damage. their key findings are:

- In response to irradiation, RIG-I is associated with chromatin
- Overexpression or activation of RIG-I suppresses NHEJ DNA repair
- Overexpressed RIG-1 interacts with XRCC4 in an RNA-dependent fashion
- RIG-I interacts with the CTD domain of XRCC4
- XRCC4 interacts with the CTD of RIG1
- Overexpressed RIG1 lowers survival upon IR, and is epistatic with XRCC4 inactivation
- RIG-I overexpression limits viral integration
- XRCC4 inactivation interferes with inflammatory responses in response to RIG-1 activation
- XRCC4 is required for RIG1 complex formation with RIG1 and MAVS, and impacts RIG-I ubiquitylation.

Overall, the observation are interesting, and point at a complex interplay between DNA repair and RNA sensing. What I miss to fully grasp the impact of this manuscript, is the physiological role of RIG-I in preventing viral integration. Overexpression of RIG-I or supraphysiological activation of RIG-I clearly block NHEJ and integration of a lentivirus. However, whether endogenous RIG-I can also prevent viral integration using the proposed mechanisms remains elusive.

Specific points:

Figure 1: I understand the rationale to use a chemical agonist to activate RIG-I, but it remains unclear to me why a RIG-I agonist is required to see effects. Figure 1A,B suggest that extensive amounts of RIG-I are present. However, IN 1G, agonist treatment is required to show expression, and the effects of sole RIG-I depletion on NHEJ is not shown. Is this a phenotype that only becomes visible upon overexpression (1F) or agonist treatment?

It is unclear if these methods to overexpress/hyperactivate RIG-I reflect physiological conditions.

Figure 1: images of 1I do not reflect the quantifications in 1J

Unclear how the RNA-dependent interaction if RIG-1 and XRCC4 is mediated, and how that fits in a model. Is RNA binding required to activate/induce complex formation of RIG-1? Is MAVS required for repression of NHEJ/viral integration?

Does 3p-hpRNA treatment cause DNA damage? important to check this

Figure 2B Panel B suggest that at endogenous levels, RIG-I interacts with XRCC4. Effects of DNA repair should likely also be visible under such conditions.

Figure 4A: these essays should also be done with shRIG-I (or even better crispr KO for RIG-I) to see effects of endogenous RIG-I. Same for panel 4D

Reviewer #3 (Remarks to the Author):

Guo et al. showed that RIG-I is recruited to DNA double strand breaks to inhibit non-homologous end joining, which is one of two DNA repair mechanisms. This inhibition is mediated by the interaction of RIG-I with XRCC4, one of the components that form the non-homologous end joining-mediating complex. The authors proposed that during retrovirus infection the inhibitory effect of RIG-I on non-homologous end joining impairs retrovirus integration into the host genome. Conversely, XRCC4 promotes RIG-I signaling by enhancing oligomerization and ubiquitination of RIG-I, thus enhancing anti-viral type I IFN responses.

The proposed concept that XRCC4 is involved in RIG-I mediated inhibition of non-homologous end joining and at the same time in the enhancement of RIG-I signaling and the induction of protective type I IFN responses is innovative and attractive. However, the study is written in a manner that it is very difficult to understand. During the whole study, the authors keep a RIG-I-centric view, which impedes descriptions of the performed experiments tremendously. Also, the experiments are described in a too minimalistic manner. Mostly, it remains unclear how often single experiments were performed and what kind of replicates were used to calculate error bars. Also, in several cases no statistics was provided. Many experiments rely on overexpression or down-modulation of selected components in cell lines. However, by using CRISPR/Cas9 gene editing also cell lines devoid of certain genes could have been generated. Furthermore, several observations could have been validated by immunocytochemistry and microscopy. The decision to base the entire study on work with cell lines limits the potential to draw relevant conclusions for in vivo conditions. Certainly, studies in cell lines are suitable to highlight some major effects in highly reproducible systems. However, in particular the more complicated read outs such as cell survival as well as infection experiments require more sophisticated systems.

Specific comments:

How exactly was the cell survival experiments in Fig. 4A and B carried out? More detailed descriptions are needed to be able to understand how exactly the colony formation assay was carried out. What values have been used to calculate error bars? Have technical replicates been tested, and if so how many? Are differences in the graphs statistically significant? This similarly applies for Fig. 4C.

In Fig. 4D-F a GFP+ lentivirus was used, whereas it is remains unclear which virus exactly was utilized. Did 3p-hpRNA treatment or transfection affect the infection efficacy of HEK293T cells? If this was the case, it would not be surprising to observe reduced integration of lentiviral DNA into the cellular genome. These experiments have to be better controlled in order to be able to draw relevant conclusions.

We thank the reviewers for the detailed and constructive comments. All the suggestions have helped us to greatly improve our manuscript.

Here are the detailed responses to reviewers' comments.

Reviewer #1 (Remarks to the Author):

In this manuscript entitled “Reciprocal regulation of RIG-I and XRCC4 connects DNA repair with RIG-I immune signaling” described the connection between immune signaling and double-stranded break (DSB) repair. They found that RIG-I, a key cytosolic RNA sensor for immune signaling, interacts with the NHEJ key factor XRCC4. This interaction impedes the interaction of XRCC4 with other NHEJ proteins, and thus suppresses NHEJ, which is required for retrovirus integration. In the other hand, XRCC4 contributes RIG-I immune signaling. Thus, these findings reveal the mechanism how RIG-I and NHEJ coordinate to inhibit virus infection.

Overall, this is a very interesting story and contains a lot of high quality data. Thus, I am exceptionally supportive of this work and recommend its publication in Nature Communications.

We would like to thank the reviewer for the support and constructive feedback. This is much appreciated.

Some comments:

1. Figure 1A-D, RIG-I is recruited to DNA damage sites using fractionation and CHIP. Can this be confirmed by IR- or Laser-induced foci.

Thanks for the suggestion. In the revised manuscript, we examined the recruitment of RIG-I to laser-induced DNA damage sites. As shown in **Fig. S1c**, RIG-I was recruited to laser-induced DNA damage sites following micro-irradiation. In addition, we utilized a reporter system in U2OS cells to induce the double-strand break (DSB) by FokI to examine the localization of RIG-I¹. Upon induction of the DSB, we found that RIG-I localized to the site of damage (**Fig. 1d**). These results are consistent with the fractionation and CHIP assay (**Fig. 1b, 1c, S1a**), suggesting that RIG-I is recruited to DNA damage sites following DNA damage.

2. The NHEJ reporter system used in this paper is very crude. The results will strongly effected by many factors, such as transfection efficiency, cell status after lacking essential gene, and so on. Therefore, a stringent assay for NHEJ is necessary, such as class switch recombination (CSR) analysis, telomere fusion analysis in TRF2-depleted cells, or foreign DNA integration analysis.

Thanks for the suggestion. We examined the effect of RIG-I on class switch recombination. Control and RIG-I knockdown CH12F3-2a cells were stimulated with ligands (CIT, TGF-β1, IL-4, and CD40 ligand), and class switch from IgM to IgA was analyzed². As shown in **Fig. 1i, S1q**, compared with control cells, RIG-I knockdown cells showed an increase in class switch efficiency, indicating that RIG-I depletion enhances NHEJ.

1i

S1q

In addition, we performed an NHEJ linearized plasmid assay that the linearized plasmid pEYFP-N1 was transfected into control and RIG-I knockdown or knockout HEK293T cells. The cells were then harvested and lysed to isolate transfected plasmids. The efficiency of end-joining repair was assessed by qPCR of the ligated YFP region, normalized to an uncut flanking DNA sequence³. As shown in **Fig. 1h, S4m**, RIG-I depletion led to an increase of end-joining repair efficiency in cells compared with that in control cells. In contrast, overexpression of RIG-I inhibited end-joining repair in cells (**Fig. 1f**). These results are consistent with the NHEJ reporter assay that measures repair of an I-SceI-induced DSB by GFP expression (**Fig. 1e, 1g**), suggesting that RIG-I plays an important role in regulating NHEJ.

1h

S4m

1f

3. The stability of LIG4 depends on its interaction with XRCC4. When RIG-I depletes XRCC4 from LIG4 after it's overexpressed or cells are treated by RIG-I agonist, is the protein level of XRCC4 reduced?

Yes. As shown in **Fig. 3a, 3b, S3a, S3b**, lower levels of LIG4 were observed in RIG-I overexpressing or 3p-hpRNA treated cells in which the interaction of LIG4 with XRCC4 was hindered by high expression of RIG-I.

4. Where is the level of non-phosphorylated IRF3 in Figure S5B-D, S5F-H?

We added the protein levels of IRF3 in the revised manuscript.

Reviewer #2 (Remarks to the Author):

In their manuscript, Guo and co-workers study the role of RIG-I on the cellular response to DNA damage. their key findings are:

- In response to irradiation, RIG-I is associated with chromatin
- Overexpression or activation of RIG-I suppresses NHEJ DNA repair
- Overexpressed RIG-1 interacts with XRCC4 in an RNA-dependent fashion
- RIG-I interacts with the CTD domain of XRCC4
- XRCC4 interacts with the CTD of RIG1
- Overexpressed RIG1 lowers survival upon IR, and is epistatic with XRCC4 inactivation
- RIG-I overexpression limits viral integration
- XRCC4 inactivation interferes with inflammatory responses in response to RIG-1 activation
- XRCC4 is required for RIG1 complex formation with RIG1 and MAVS, and impacts RIG-I ubiquitylation.

Overall, the observations are interesting, and point to a complex interplay between DNA repair and RNA sensing. What I miss to fully grasp the impact of this manuscript, is the physiological role of RIG-I in preventing viral integration. Overexpression of RIG-I or supraphysiological activation of RIG-I clearly blocks NHEJ and integration of a lentivirus. However, whether endogenous RIG-I can also prevent viral integration using the proposed mechanisms remains elusive.

Thank you for the terrific comments and suggestions, which were critical to the advancement of our study.

Specific points:

Figure 1: I understand the rationale to use a chemical agonist to activate RIG-I, but it remains unclear to me why a RIG-I agonist is required to see effects. Figure 1A,B suggest that extensive amounts of RIG-I are present. However, in 1G, agonist treatment is required to show expression, and the effects of sole RIG-I depletion on NHEJ is not shown. Is this a phenotype that only becomes visible upon overexpression (1F) or agonist treatment?

It is unclear if these methods to overexpress/hyperactivate RIG-I reflect physiological conditions.

In the revised manuscript, we examined the effect of endogenous RIG-I on NHEJ repair. Firstly, we employed the NHEJ reporter assay that measures repair of an I-SceI-induced DSB by GFP expression. As shown in **Fig. S1o, 1g, S4l**, RIG-I knockdown or knockout led to an increase of NHEJ efficiency. Then, we performed an NHEJ linearized plasmid assay that the linearized plasmid pEYFP-N1 was transfected into control and RIG-I knockdown or knockout HEK293T cells. The efficiency of end-joining repair was assessed by qPCR of the ligated YFP region, normalized to an uncut flanking DNA sequence. As shown in **Fig. 1h, S4m**, RIG-I depletion led to an increase of end-joining repair efficiency in cells compared with that in control cells. Furthermore, we assessed the effect of endogenous RIG-I on class switch recombination. Control and RIG-I knockdown CH12F3-2a cells were stimulated with ligands (CIT, TGF- β 1, IL-4, and CD40 ligand), and class switch from IgM to IgA was analyzed. As shown in **Fig. 1i**, compared with control cells, RIG-I knockdown cells showed an increase in class switch efficiency, indicating that RIG-I depletion enhanced NHEJ. These results suggest that RIG-I plays an important role in regulating NHEJ.

S1o

Figure 1: images of 1I do not reflect the quantifications in 1J

We repeated the experiment and quantified γ H2AX foci. As shown in **Fig. 1l**, **S1r**, more γ H2AX foci were observed in RIG-I overexpressing cells at late time points after IR treatment, suggesting RIG-I suppresses DNA repair. For the quantification of γ H2AX foci, each dot represents a single cell, and 100 cells were counted in each group for this experiment. Error bars represent SEM from this experiment (**Fig. 1l**).

S1r

11

Unclear how the RNA-dependent interaction if RIG-1 and XRCC4 is mediated, and how that fits in a model. Is RNA binding required to activate/induce complex formation of RIG-1? Is MAVS required for repression of NHEJ/viral integration?

To test whether RNA binding is required for the complex formation of RIG-I and XRCC4 in response to DNA damage, we generated two RIG-I mutants that are defective in binding RNA (K858, 861A; T347A)^{4,5}, and examined their interaction with XRCC4. As shown in **Fig. S2b, S2c, S2e**, the interaction of wild type (WT), but not RNA binding-deficient RIG-I mutants, with XRCC4 was increased following IR treatment. Consistently, overexpression of WT, but not RNA binding-deficient RIG-I mutants, suppressed NHEJ, suggesting that RNA binding is required for RIG-I to interact with XRCC4 following DNA damage, thereby suppressing NHEJ.

S2b

S2c

S2e

In addition, we examined whether MAVS, which is indispensable for RIG-I immune signaling, is required for RIG-I mediated suppression of NHEJ and viral integration. As shown in **Fig. S2f, S2g, S4r** RIG-I overexpression inhibited NHEJ and viral integration in both control and MAVS knockdown cells. These results suggest that RIG-I suppressed NHEJ through interacting with XRCC4 in a RNA-dependent manner, while MAVS is dispensable for RIG-I mediated inhibition of NHEJ.

Does 3p-hpRNA treatment cause DNA damage? important to examine this

We examined the effect of 3p-hpRNA on DNA damage response and repair. U2OS cells were treated with 3p-hpRNA followed by irradiation (IR) treatment, and γ H2AX foci, markers of DSB, were detected by immunofluorescence. As shown in **Fig. S1s, S1t**, in the absence of IR treatment, 3p-hpRNA treatment did not cause obvious DNA damage in cells. However, upon treatment with IR, more γ H2AX foci were observed in 3p-hpRNA treated cells at late time points (8h and 24h), which suggests that 3p-hpRNA treatment inhibits DNA repair.

Figure 2B Panel B suggest that at endogenous levels, RIG-I interacts with XRCC4. Effects of DNA repair should likely also be visible under such conditions.

In the revised manuscript, we examined the effect of endogenous RIG-I on DNA damage response and repair. Control and RIG-I knockdown or knockout cells (U2OS, A549) were treated with IR and γ H2AX foci, markers of DSB, were detected by immunofluorescence. As shown in **Fig. 1j, 1k, S4j, S4k, S4n, S4o**, compared with control cells, RIG-I depleted cells showed a decrease of γ H2AX foci at the late time point (8h), which suggests that RIG-I depletion promotes DNA repair.

Figure 4A: these essays should also be done with shRIG-I (or even better crispr KO for RIG-I) to see effects of endogenous RIG-I. Same for panel 4D

Thanks for the suggestion. In the revised manuscript, we examined the effect of endogenous RIG-I on cellular sensitivity to irradiation (IR). Control and RIG-I knockdown or knockout A549 cells were treated with IR, and cell survival was analyzed by colony formation assay. As shown in **Fig. 4c, 4d**, RIG-I depletion rendered cells more resistant to IR treatment. Furthermore, we examined the effect of RIG-I on cellular sensitivity to IR in vivo. As shown in **Fig. 4e, S4p**, RIG-I depletion rendered tumors more resistant to IR in the xenograft model. These results suggest that loss of RIG-I rendered cells more resistant to IR through promotion of NHEJ (**Fig. 1g-1i, S4l-4m**).

4c**4d****S4p****4e**
In addition, we examined the effect of endogenous RIG-I on viral integration. We generated RIG-I knockdown MEF cells, and evaluated the effect of RIG-I on viral integration. As shown in **Fig. 4i-4k**, higher genomic GFP DNA levels and more provirus copies were observed in RIG-I knockdown cells than that in control cells. In addition, WT and RIG-I knockout HEK293T cells were infected with GFP positive lentiviruses, and viral integration was evaluated by analyzing GFP DNA levels in the genome. As shown in **Fig. S4x**, higher genomic GFP DNA levels were observed in RIG-I knockout cells compared with that in WT cells, suggesting that loss of RIG-I promotes viral integration into the host genome.

4i**4j****4k****S4x****Reviewer #3 (Remarks to the Author):**

Guo et al. showed that RIG-I is recruited to DNA double strand breaks to inhibit non-homologous end joining, which is one of two DNA repair mechanisms. This inhibition is mediated by the interaction of RIG-I with XRCC4, one of the components that form the non-homologous end joining-mediating complex. The authors proposed that during retrovirus infection the inhibitory effect of RIG-I on non-homologous end joining impairs retrovirus integration into the host genome. Conversely, XRCC4 promotes RIG-I signaling by enhancing oligomerization and ubiquitination of RIG-I, thus enhancing anti-viral type I IFN responses.

The proposed concept that XRCC4 is involved in RIG-I mediated inhibition of non-homologous end joining and at the same time in the enhancement of RIG-I signaling and the induction of protective type I IFN responses is innovative and attractive. However, the study is written in a manner that it is very difficult to understand. During the whole study, the authors keep a RIG-I-centric view, which impedes descriptions of the performed experiments tremendously. Also, the experiments are described in a too minimalistic manner. Mostly, it remains unclear how often single experiments were performed and what kind of replicates were used to calculate error bars. Also, in several cases no statistics was provided. Many experiments rely on overexpression or down-modulation of selected components in cell lines. However, by using CRISPR/Cas9 gene editing also cell lines devoid of certain genes could have been generated. Furthermore, several observations could have been validated by immunocytochemistry and microscopy. The decision to base the entire study on work with cell lines limits the potential to draw relevant conclusions for in vivo conditions.

Certainly, studies in cell lines are suitable to highlight some major effects in highly reproducible systems. However, in particular the more complicated read outs such as cell survival as well as infection experiments require more sophisticated systems.

Thank you for the detailed and constructive comments, which helped us to greatly improve the quality of our manuscript.

(1) In the revised manuscript, we examined the recruitment of RIG-I to laser-induced DNA damage sites. As shown in **Fig. S1c**, RIG-I was recruited to laser-induced DNA damage sites following micro-irradiation. In addition, we utilized a reporter system in U2OS cells to induce the double-strand break (DSB) by FokI to examine the localization of RIG-I. Upon induction of the DSB, we found that RIG-I localized to the site of damage (**Fig. 1d**). These results are consistent with the fractionation and CHIP assay (**Fig. 1b, 1c, S1a**), suggesting that RIG-I is recruited to DNA damage sites following DNA damage.

S1c

1d

(2) We examined the effect of endogenous RIG-I on cellular sensitivity to irradiation *in vitro* and *in vivo*. Control and RIG-I knockdown or knockout A549 cells were treated with IR, and cell survival was analyzed by colony formation assay. As shown in **Fig. 4c, 4d**, RIG-I depletion rendered cells more resistant to IR treatment. Furthermore, we examined the effect of RIG-I on cellular sensitivity to IR *in vivo*. As shown in **Fig. 4e, S4p**, RIG-I depletion rendered tumors more resistant to IR in the xenograft model. These results suggest that loss of RIG-I rendered cells more resistant to IR, consistent with RIG-I's role in inhibiting NHEJ (**Fig. 1g-1i, S4l-4m**).

4c

4d

S4p

4e

(3) We examined the effect of endogenous RIG-I on viral integration. We generated RIG-I knockdown MEF cells, and evaluated the effect of RIG-I on viral integration. As shown in **Fig. 4i-4k**, higher genomic GFP DNA levels and more provirus copies were observed in RIG-I knockdown cells than that in control cells. In addition, WT and RIG-I knockout HEK293T cells were infected with GFP positive lentiviruses, and viral integration was evaluated by analyzing GFP DNA levels in the genome. As shown in **Fig. S4x**, higher genomic GFP DNA levels were observed in RIG-I knockout cells compared with that in WT cells, suggesting that loss of RIG-I promoted viral integration into the host genome.

(4) We examined the role of XRCC4 in suppressing influenza virus replication *in vivo*. C57BL/6 mice were inhaled aerosolized control or XRCC4 siRNA mixed with *in vivo*-jetPEI® using an air-compressing nebulizer (at an estimated inhalable dose of 3mg/kg)⁶⁻¹⁰. After 24 h, mice were inoculated intranasally with 10³ PFU of influenza virus A/WSN/1933. Further treatment of siRNA was performed at 24 and 48 h after virus infection respectively. As shown in **Fig. S6d**, XRCC4 siRNA downregulated XRCC4 expression in lungs of mice compared with that in lungs of control mice. Lung XRCC4 knockdown mice displayed faster body weight loss and poorer survival than control mice during viral infection (**Fig. 6e-6f**). In addition, lung XRCC4 knockdown mice exhibited a greater degree of lung injury caused by influenza virus infection than control mice (**Fig. 6g, S6e**), suggesting that silencing XRCC4 in lung promotes influenza virus replication *in vivo*. Consistent with these observations, higher viral NP RNA levels and lower IFN beta RNA levels were observed in lung tissues with XRCC4 knockdown, compared with that in control tissues (**Fig. 6h, S6f**). These results indicate that XRCC4 depletion renders mice more susceptible to influenza virus infection.

6e**6g****6f****6h****S6d****S6e****S6f**
(5) In the revised manuscript, we described the experiments in detail in the figure legends.

Specific comments:

How exactly was the cell survival experiments in Fig. 4A and B carried out? More detailed descriptions are needed to be able to understand how exactly the colony formation assay was carried out. What values have been used to calculate error bars? Have technical replicates been tested, and if so how many? Are differences in the graphs statistically significant? This similarly applies for Fig. 4C.

A549 cells overexpressing RIG-I (**Fig. 4a**), RIG-I and XRCC4 shRNA (**Fig. 4b**), or RIG-I and IFNAR2 shRNA (**Fig. S4d**), were seeded in triplicated wells of 6-well plates and then treated with irradiation (IR) as indicated. After incubated for 12-14 days, colonies were stained and counted. The survival of untreated cells was normalized to 100%. Data shown are representative of three independent experiments and values are mean \pm SEM of technical replicates ($n=3$). Another representative of three independent experiments is shown in **Fig. S4a, S4b** and **S4e** respectively. In the revised manuscript, we described in detail in the figure legends.

In Fig. 4D-F a GFP+ lentivirus was used, whereas it is remains unclear which virus exactly was utilized. Did 3p-hpRNA treatment or transfection affect the infection efficacy of HEK293T cells? If this was the case, it would not be surprising to observe reduced integration of lentiviral DNA into the cellular genome. These experiments have to be better controlled in order to be able to draw relevant conclusions. The GFP+ lentivirus was packaged in HEK293T cells. The lentiviral vector expressing GFP, and packaging constructs (pMD2.G and pSPAX2) were co-transfected into HEK293T cells. Media containing lentiviruses was collected 48 hours after transfection. Harvested media was added to infect cells for further experiments. Equal number of cells were infected with equal viruses for quantifying the viral integration into host genome.

We evaluated the viral infection efficiency by quantifying GFP RNA levels in cells during early infection with the lentivirus (2 h). As shown in **Fig. S4u**, comparable GFP RNA levels were observed in 3p-hpRNA treated and untreated cells. In addition, neither RIG-I overexpression nor depletion had a significant effect on viral infection into host cells (**Fig. S4v, S4w**). These results suggest that it is the viral integration into host genome but not viral infection into host cells that was regulated by RIG-I.

References

- 1 Shanbhag, N. M. & Greenberg, R. A. The dynamics of DNA damage repair and transcription. *Methods in molecular biology* **1042**, 227-235, doi:10.1007/978-1-62703-526-2_16 (2013).

- 2 Nowsheen, S. *et al.* L3MBTL2 orchestrates ubiquitin signalling by dictating the sequential recruitment of RNF8 and RNF168 after DNA damage. **20**, 455-464, doi:10.1038/s41556-018-0071-x (2018).
- 3 Zhang, Q. *et al.* FBXW7 Facilitates Nonhomologous End-Joining via K63-Linked Polyubiquitylation of XRCC4. *Molecular cell* **61**, 419-433, doi:10.1016/j.molcel.2015.12.010 (2016).
- 4 Lassig, C. *et al.* ATP hydrolysis by the viral RNA sensor RIG-I prevents unintentional recognition of self-RNA. *eLife* **4**, doi:10.7554/eLife.10859 (2015).
- 5 Nabet, B. Y. *et al.* Exosome RNA Unshielding Couples Stromal Activation to Pattern Recognition Receptor Signaling in Cancer. *Cell* **170**, 352-366 e313, doi:10.1016/j.cell.2017.06.031 (2017).
- 6 Li, F. *et al.* Robust expression of vault RNAs induced by influenza A virus plays a critical role in suppression of PKR-mediated innate immunity. *Nucleic acids research* **43**, 10321-10337, doi:10.1093/nar/gkv1078 (2015).
- 7 Besch, R. *et al.* Proapoptotic signaling induced by RIG-I and MDA-5 results in type I interferon-independent apoptosis in human melanoma cells. *The Journal of clinical investigation* **119**, 2399-2411, doi:10.1172/jci37155 (2009).
- 8 Bitko, V., Musiyenko, A., Shulyayeva, O. & Barik, S. Inhibition of respiratory viruses by nasally administered siRNA. *Nature medicine* **11**, 50-55, doi:10.1038/nm1164 (2005).
- 9 Campbell, M. *et al.* Targeted suppression of claudin-5 decreases cerebral oedema and improves cognitive outcome following traumatic brain injury. *Nature communications* **3**, 849, doi:10.1038/ncomms1852 (2012).
- 10 Zafra, M. P. *et al.* Gene silencing of SOCS3 by siRNA intranasal delivery inhibits asthma phenotype in mice. *PloS one* **9**, e91996, doi:10.1371/journal.pone.0091996 (2014).

REVIEWERS' COMMENTS

Reviewer #1 (Remarks to the Author):

All my concerns have been addressed.

Reviewer #2 (Remarks to the Author):

the authors answered my comments adequately and I support publication of this manuscript. one things that I noticed in the revised manuscript is the laser stripe experiment in S1c. the authors conclude that Rig-1 localises to laser-induced breaks. However, I hardly see any signal there. the authors should quantify these images, other the data does not really support the conclusion and the conclusion should be toned down.

Reviewer #3 (Remarks to the Author):

The authors provided a lengthy point by point reply and added a lot of additional information to the manuscript. They showed the newly presented data not only in the revised manuscript but also in the point by point reply, in which they also copy pasted the passages that they wrote for the revision of the manuscript. This way, many of the questions were not directly answered by the authors, whereas instead the reviewers had to reread the questions, the manuscript and the answers in order to see how the replies would fit. In particular, the answers to reviewer #3 did not directly address the formulated concerns. Instead, the reply (1) was copy pasted from the reply to reviewer #1, whereas the replies (2) and (3) were copy pasted from the reply to reviewer #2, despite the corresponding concerns by far were not the identical ones. This way of handling reviewer comments is rather unusual and not necessarily helpful to specifically address quality deficits of the original manuscript.

Nevertheless, reply (4) to comments of reviewer #3 was formulated specifically in reply to one comment of reviewer #3. In this part, new and very important data were provided, i.e. the in vivo analysis of the effect of XRCC4 down modulation on influenza A virus infection. The presented data clearly indicate that under conditions of XRCC4 down-modulation mice show enhanced sensitivity to lethal influenza A virus infection. Although these data look very good, the authors did not reformulate neither the abstract nor the discussion. This is a pity, because these both parts now are immature, when considering the vast amount of new data included in the revised manuscript.

It is appreciated that the authors provided a more detailed description of the cell survival experiment, which indeed facilitated understanding the experiment that was performed in Fig. 4A and B. Furthermore, some additional descriptions were given for the lentivirus that was used in Fig. 4D-F.

The new control data shown in Fig. S4U-W of the revised manuscript indeed show that neither 3p-hpRNA treatment nor transfection with shRIG-I directly affected the expression of GFP, which was tested as a read out of virus infection. Indeed, these data suggest that RIG-I signaling does not directly affect virus infection of the cells. I agree with the authors that correspondingly one potential explanation of the data shown in Fig. 4D-F is that RIG-I signaling impaired viral integration into the host genome. However, for me it is still difficult to understand why the authors did not try to perform other experiments to more directly address this hypothesis. Considering the relevance of the conclusion, more direct experimental evidence would have been particularly appreciated.

In the figure legends, the authors provided more detailed descriptions for the single experiments. They also revealed group sizes in all Figure legends. In the legend of Fig. 4 the even reported how often the respective experiments were repeated (see legend of Fig. 4D). However, it remains largely unclear how often the other experiments in the manuscript have been repeated. Therefore, estimation of the significance of the shown phenomena is still difficult.

Overall, the concerns of all three reviewers were largely addressed. Although the authors added some additional information on how the experiments were performed, they missed the opportunity to provide in depth information about their experiments. Correspondingly, the shown data are difficult evaluate. Despite the authors added significant amounts of new data, they did not adapt neither the abstract nor the discussion. Whereas the readability of the results section significantly improved, the abstract is still difficult to understand and the discussion is rather premature.

REVIEWERS' COMMENTS

We thank the reviewers for the positive comments. This is much appreciated.

Reviewer #1 (Remarks to the Author):

All my concerns have been addressed.

We thank the reviewer for the positive comments.

Reviewer #2 (Remarks to the Author):

the authors answered my comments adequately and I support publication of this manuscript.

one things that I noticed in the revised manuscript is the laser stripe experiment in S1c. the authors conclude that Rig-1 localises to laser-induced breaks. However, I hardly see any signal there. the authors should quantify these images, other the data does not really support the conclusion and the conclusion should be toned down.

We thank the reviewer for the positive comments. In the revised manuscript, we provided better representative images in **Fig. S1c**.

S1c

Reviewer #3 (Remarks to the Author):

The authors provided a lengthy point by point reply and added a lot of additional information to the manuscript. They showed the newly presented data not only in the revised manuscript but also in the point by point reply, in which they also copy pasted the passages that they wrote for the revision of the manuscript. This way, many of the questions were not directly answered by the authors, whereas instead the reviewers had to reread the questions, the manuscript and the answers in order to see how the replies would fit. In particular, the answers to reviewer #3 did not directly address the formulated concerns. Instead, the reply (1) was copy pasted from the reply to reviewer #1, whereas the replies (2) and (3) were copy pasted from the reply to reviewer #2, despite the corresponding concerns by far were not the

identical ones. This way of handling reviewer comments is rather unusual and not necessarily helpful to specifically address quality deficits of the original manuscript.

We are sorry some of the responses did not directly address the reviewer's comment. We reorganized our response as below.

(1) The reviewer suggested that "Several observations could have been validated by immunocytochemistry and microscopy." In the original manuscript, we examined the recruitment of RIG-I to DNA damage sites using fractionation and CHIP assays (**Fig. 1b, 1c, S1a**). To further confirm the recruitment of RIG-I to damage sites, which is important for RIG-I mediated inhibition of NHEJ, we performed immunofluorescence assays. As shown in **Fig. S1c**, RIG-I was recruited to laser-induced DNA damage sites following micro-irradiation. In addition, we utilized a reporter system to induce the double-strand break (DSB) by FokI in U2OS cells. Upon induction of the DSB, we found that RIG-I localized to the site of damage (**Fig. 1d**), suggesting that RIG-I is recruited to DNA damage sites following DNA damage.

(2) We agree with the reviewer that studies in cell lines are suitable to highlight some major effects, but more sophisticated systems are needed to evaluate the effect of RIG-I on cell survival. Therefore, we examined the effect of RIG-I on cellular sensitivity to irradiation (IR) *in vivo*. Control or RIG-I knockdown A549 cells were injected subcutaneously into the flanks of 6-week-old female athymic nude mice followed by IR treatment. As shown in **Fig. 4e, S4p**, RIG-I depletion rendered tumors more resistant to IR in the xenograft model. These results suggest that loss of RIG-I rendered cells more resistant to IR, consistent with RIG-I's role in inhibiting NHEJ (**Fig. 1g-1i, S4l-4m**).

(3) The reviewer suggested CRISPR/Cas9 editing could be employed. Therefore, we generated RIG-I knockout A549 cells, and examined the effect of RIG-I on cellular sensitivity to irradiation (IR). Control and RIG-I knockout cells were treated with IR, and cell survival was analyzed by colony formation assay. As shown in **Fig. 4d**, loss of RIG-I rendered cells more resistant to IR treatment. Consistently, increased NHEJ efficiency and less γ H2AX foci were observed in RIG-I depleted cells (**Fig. S4l-4o**), suggesting that loss of RIG-I rendered cells more resistant to IR treatment through promotion of NHEJ.

In addition, we examined the effect of RIG-I on viral integration using wild type (WT) and RIG-I knockout HEK293T cells. Cells were infected with GFP positive lentiviruses, and viral integration was evaluated by analyzing GFP DNA levels in the genome. As shown in **Fig. S4x**, higher genomic GFP DNA levels were observed in RIG-I knockout cells compared with that in WT cells, suggesting that loss of RIG-I promoted viral integration into the host genome.

S4x

Nevertheless, reply (4) to comments of reviewer #3 was formulated specifically in reply to one comment of reviewer #3. In this part, new and very important data were provided, i.e. the in vivo analysis of the effect of XRCC4 down modulation on influenza A virus infection. The presented data clearly indicate that under conditions of XRCC4 down-modulation mice show enhanced sensitivity to lethal influenza A virus infection. Although these data look very good, the authors did not reformulate neither the abstract nor the discussion. This is a pity, because these both parts now are immature, when considering the vast amount of new data included in the revised manuscript.

We thank the reviewer for the positive comments. We adapted the abstract and discussion in the revised manuscript.

It is appreciated that the authors provided a more detailed description of the cell survival experiment, which indeed facilitated understanding the experiment that was performed in Fig. 4A and B. Furthermore, some additional descriptions were given for the lentivirus that was used in Fig. 4D-F.

We thank the reviewer for the positive comments.

The new control data shown in Fig. S4U-W of the revised manuscript indeed show that neither 3p-hpRNA treatment nor transfection with shRIG-I directly affected the expression of GFP, which was tested as a read out of virus infection. Indeed, these data suggest that RIG-I signaling does not directly affect virus infection of the cells. I agree with the authors that correspondingly one potential explanation of the data shown in Fig. 4D-F is that RIG-I signaling impaired viral integration into the host genome. However, for me it is still difficult to understand why the authors did not try to perform other experiments to more directly address this hypothesis. Considering the relevance of the conclusion, more direct experimental evidence would have been particularly appreciated.

We thank the reviewer for agreeing that RIG-I signaling does not directly affect virus infection of the cells. This in combination with our Fig. 4f-h, shows RIG-I expression decrease GFP gene levels in the genomic DNA in a XRCC4 dependent manner. In addition to overexpression, we knocked down/out endogenous RIG-I and determined the effect on viral integration. As shown in Fig. 4i-4j, S4x, higher genomic GFP DNA levels were observed in RIG-I knockdown MEF cells and RIG-I knockout HEK293T cells. Furthermore, we quantified provirus copies in RIG-I depleted MEF cells. As shown in Fig. 4k, more provirus copies were observed in RIG-I depleted cells, suggesting that loss of RIG-I promoted viral integration into the host genome. Overall, these observations support a critical role of RIG-I in suppressing viral integration into host genome. Although other steps from initial infection to eventual viral genome integration could still be affected by RIG-I, we think its inhibitory role on XRCC4 and NHEJ is a major factor, given the key role of NHEJ in viral integration into the genome.

In the figure legends, the authors provided more detailed descriptions for the single experiments. They also revealed group sizes in all Figure legends. In the legend of Fig. 4 the even reported how often the respective experiments were repeated (see legend of Fig. 4D). However, it remains largely unclear how often the other experiments in the manuscript have been repeated. Therefore, estimation of the significance of the shown phenomena is still difficult.

We thank the reviewer for the positive comments. In the revised manuscript, we adapted the figure legends, and provided more detailed descriptions for all experiments.

Overall, the concerns of all three reviewers were largely addressed. Although the authors added some additional information on how the experiments were performed, they missed the opportunity to provide in depth information about their experiments. Correspondingly, the shown data are difficult evaluate. Despite the authors added significant amounts of new data, they did not adapt neither the abstract nor the discussion. Whereas the readability of the results section significantly improved, the abstract is still difficult to understand and the discussion is rather premature.

In the revised manuscript, we adapted the abstract and discussion based on the newly added data. We thank the reviewer for the constructive comments to strengthen the manuscript and make it more rigorous and readable.